# STELLA: LEVERAGING STRUCTURAL REPRESENTATIONS TO ENHANCE PROTEIN UNDERSTANDING WITH MULTIMODAL LLMS

## ABSTRACT

Protein biology centers on the intricate relationships among sequence, structure, and function (text), with structure understanding being a crucial aspect for uncovering protein biological functions. Traditional methods based on protein language models (pLMs) often focus on specific aspects of biological function prediction but do not account for the broader, dynamic context of protein research—an important component for addressing the complexity of protein biology. Modern large language models (LLMs) excel in human-machine interaction, language understanding and generation, at a human-like level. By bridging structural representations with the contextual knowledge encoded within LLMs, STELLA leverages the strengths of LLMs to enable versatile and accurate predictions in protein-related tasks. It showcases the transformative potential of multimodal LLMs as a novel paradigm besides pLMs in advancing protein biology research by achieving state-of-the-art performance in both functional description and enzyme-catalyzed reaction prediction tasks. This study not only establishes an innovative LLM-based paradigm to understand proteins, but also expands the boundaries of LLM capabilities in protein biology. To foster collaboration and inspire further innovation, the codes, datasets, and pre-trained models are made publicly available at the anonymous GitHub repository `https://anonymous.4open.science/r/STELLA-DF00`.

## 1 INTRODUCTION

Protein biology revolves around the interplay of three data modalities: sequence, structure, and function (text). The principle "sequence determines structure, and structure determines function" underscores the critical link between a protein's amino acid sequence, its tertiary structure, and its biological role, such as its main functions and enzyme-catalyzed reactions. Structural data offer significant insights into how a protein's three-dimensional conformation, including features such as active sites and binding pockets, enables and regulates its core biological functions. Accurate understanding of these biological functions plays a pivotal role in advancing disease research, drug discovery, metabolic pathway analysis, and the design of enzymes for medical and biotechnological applications.

Although extensive structural data have been accumulated through decades of protein science research, including experimentally determined structures in the RCSB Protein Data Bank (PDB) [1] (Berman et al., 2000) and computationally predicted structures in the AlphaFold Protein Structure Database (AFDB) [2] (Varadi et al., 2021) by AlphaFold 2 (AF2) (Jumper et al., 2021), further efforts are needed to leverage these resources for deeper understanding of protein biological functions. The PDB, as one of the most comprehensive repositories of experimentally determined protein structures, has long served as a cornerstone of structural biology and biology computational models, such as AlphaFold 3 (Abramson et al., 2024) and ESM3 (Hayes et al., 2024). Similarly, the AFDB has dramatically increased access to high-quality predicted protein structures. These vast structural datasets provide a valuable foundation for advancing protein science, offering new opportunities to deepen our understanding of proteins. However, fully realizing their potential requires bridging

---

[1] `https://www.rcsb.org/`
[2] `https://alphafold.ebi.ac.uk/`

the gap between structural data and the functional and biochemical insights essential for practical applications in both research and industry.

Understanding structures is crucial for uncovering biological functions, such as protein functional descriptions, elucidating enzyme-catalyzed reactions, and addressing fundamental biological questions, as emphasized in this study. Previous efforts in protein function analysis have included methods such as clustering methods based on protein structure similarity (Barrio-Hernandez et al., 2023; Huang et al., 2023) and text generation methods (Abdine et al., 2023). While these approaches have contributed valuable insights, they often fall short of fully capturing the intricate and multidimensional relationships between protein structure and function, limiting their ability in addressing the complexity of protein biology. Furthermore, these methods typically lack iterative feedback mechanisms from domain experts, which are essential for aligning results with their diverse research objectives. Predicting enzyme-catalyzed reactions is another complex task in protein science, attracting significant research attention (Derevyanko et al., 2018; Steinegger et al., 2019; Hermosilla et al., 2021; Zhang et al., 2022; Hermosilla and Ropinski, 2022; Fan et al., 2022). Although progress has been made, existing methods often approach enzyme prediction as a multi-label classification task, which still struggles with accurately predicting enzyme classes that have not been thoroughly explored in high-throughput proteomics studies. These limitations hinder the ability to fully understand the functions of such enzymes, underscoring the need for more advanced approaches to achieve reliable predictions and accelerate research in this area.

To address this challenge, innovative approaches that integrate structural data with cutting-edge computational tools are urgently needed. Recent advancements, including Prot2Text (Abdine et al., 2023), ProteinGPT (Xiao et al., 2024), and ProtChatGPT (Wang et al., 2024a), have explored the utilization of multimodal LLMs in protein biology. These models typically integrate protein sequence and structure data using a late fusion strategy, where each modality is encoded separately before being aligned or combined. However, late fusion approaches have certain limitations, such as the potential loss of cross-modal relationships and increased complexity of encoder modules. In contrast, the early fusion strategy—where different modalities are jointly represented and fused into a unified representation at encoding stage—has the potential to both preserve the intrinsic relationships between modalities and improve computational efficiency. Motivated by the aforementioned perspectives, this work investigates the advantages of early fusion for multimodal LLMs modeling in protein biology.

To leverage the potential of multimodal LLMs with an early fusion strategy in protein biology, this study introduces STELLA, a multimodal LLM designed to bridge protein language and natural language, enabling the learning of complex structure-function relationships from multimodal data. Unlike previous approaches that use late fusion strategies, STELLA utilizes ESM3 encoder (esm3_sm_open_v1) (Hayes et al., 2024), which inherently implements an early fusion mechanism, where protein sequence and structure are jointly represented in a unified encoding process. By leveraging these fused structural representations—integrating both sequence and structural information—STELLA enhances protein understanding through the power of LLMs, enabling it to interpret protein tertiary structures and predict functional descriptions and enzyme-catalyzed reactions from diverse and versatile user prompts. Apart from the advancement of protein language models (pLMs), STELLA highlights the transformative potential of multimodal LLMs in advancing protein biology research by achieving state-of-the-art performance in both tasks. In doing so, it offers a new paradigm for understanding proteins and extends the capabilities of general-purpose LLMs in the field of protein biology. The key contributions of this study include:

**1.** By inheriting the early fusion mechanism of ESM3, STELLA achieves state-of-the-art performance in protein functional description and enzyme-catalyzed reaction prediction tasks.

**2.** This study constructs a large-scale multimodal instruction tuning dataset, OPI-Struc, to support training of multimodal LLMs for protein-related tasks.

**3.** This study presents the methodology, architecture, and performance of STELLA, alongside the open source code, data, and pre-trained models to encourage collaboration and further advancements in the field.

We anticipate that this study will help drive the advancement of protein science and computational biology through LLM-based approaches, establishing a new paradigm beyond the pLM-based paradigms.

## 2 RELATED WORK

### 2.1 PROTEIN-TEXT MODELING

The long-term goal of protein representation learning is to extract biologically relevant information from diverse data modalities, including amino acid sequences and tertiary structures (i.e., protein language) as well as relevant texts in natural language that encapsulate protein related knowledge. Aligning the protein language and natural language has emerged as a crucial aspect of advancing protein representation learning, and attracted much attention in the research community. For instances, ProtST (Xu et al., 2023) utilizes contrastive learning to align amino acid sequences with biomedical texts, aiming to obtain biologically informative protein embeddings that can be applied to various downstream tasks. Besides protein representation learning, ProteinDT (Liu et al., 2023a) leverages textual data to enhance protein design in text-to-sequence generation tasks. Additionally, Prot2Text (Abdine et al., 2023) proposes a method of aligning protein structures and function description texts by using a fused multimodal encoder-decoder framework. In Prot2Text, the encoder is composed of a Relational Graph Convolutional Neural Network (RGCN) for encoding protein structures and a ESM2-35M (Lin et al., 2022) for encoding amino acid sequences and the decoder is a pretrained GPT-2 model to generate protein function annotations. Before the prevalence of LLMs, protein representation learning mainly focuses on single modality like amino acid sequences, or sequence-text alignment by contrastive learning. Hardly any research engages in how to effectively bridge biological language (e.g., protein tertiary structures) to the massive knowledge embedded in natural language that plays a pivotal role in both scientific communication and discovery. As we all know, the process of scientific discovery is a procedure propelled by communication among domain experts and iterative experimentation. Therefore, the excellent conversation and reasoning abilities of LLMs are highly expected to empower the process of scientific discovery.

### 2.2 LLMs FOR PROTEIN BIOLOGY

Recent studies have highlighted the potential of LLMs in advancing biomedical research, spanning molecules, proteins, and RNA. In the specific domain of protein biology, several notable developments have emerged. ProTokens (Lin et al., 2023) employs discrete and compressed protein tokens that encode rich structural information for LLMs. These tokens are learned through an autoencoder framework, with both the input and output consisting of 3D protein structures. InstructProtein(Wang et al., 2023) constructs instruction datasets derived from a knowledge graph to address the annotation imbalance present in previous protein-text datasets. This dataset is utilized to fine-tune LLMs for aligning protein sequences with natural language, enabling bidirectional tasks such as predicting functions from sequences and generating protein sequences from natural language prompts. BioMedGPT (Luo et al., 2023) employs a fully-connected layer to connect an amino acid sequence encoder, ESM-2-3B (Lin et al., 2022), and Llama2-Chat-7B (Touvron et al., 2023), which has been incrementally pretrained on biomedical literature from S2ORC (Lo et al., 2020). ProteinChat (Huo et al., 2024) represents a more recent multi-modal LLM designed to predict protein functions. It integrates a protein sequence encoder, xTrimoPGLM (Chen et al., 2024), with the Vicuna-13B model (Zheng et al., 2023) through a linear layer adapter. Trained on over 1.5 million protein-related (protein, prompt, answer) triplets from the Swiss-Prot dataset, ProteinChat covers a wide range of protein functions. By taking an amino acid sequence as input, it generates comprehensive narratives detailing the functional properties of the given protein.

## 3 A FIRST LOOK AT STELLA'S CAPABILITIES THROUGH CASE STUDIES

STELLA demonstrates outstanding performance in protein understanding by integrating structural representations into LLMs. As illustrated in Figure 1 (left), STELLA excels in following natural language instructions and providing responses that align with the research goals of human specialists. During the interaction, STELLA correctly identified the primary function of the newly reviewed protein G1TFE0 in the Swiss-Prot database, accurately recognizing it as a component of the large ribosomal subunit. As the dialogue progressed, STELLA elaborated on the core constituents of the ribonucleoprotein complex, highlighting its extensive domain knowledge. Additionally, STELLA showcased its reasoning capabilities by linking the loss of ribosomal function to cellular dysfunctions, demonstrating its ability to establish connections between complex biological processes. In the right

panel of Figure 1, STELLA accurately predicted the function of another newly characterized protein, A0A1D0BR98. Upon further inquiry from the user, it explained details about the mechanisms of the toxin and provided practical treatments. Both case studies illustrate STELLA's ability to predict protein functions from structural data and to deliver informative, contextually relevant responses tailored to diverse research objectives. Moreover, STELLA's robust reasoning abilities enable it to assist domain experts in uncovering connections between protein functions and disease mechanisms, further highlighting its potential to drive scientific discovery and innovation. More example demonstrations are shown in Figure 4 of Appendix A.1.

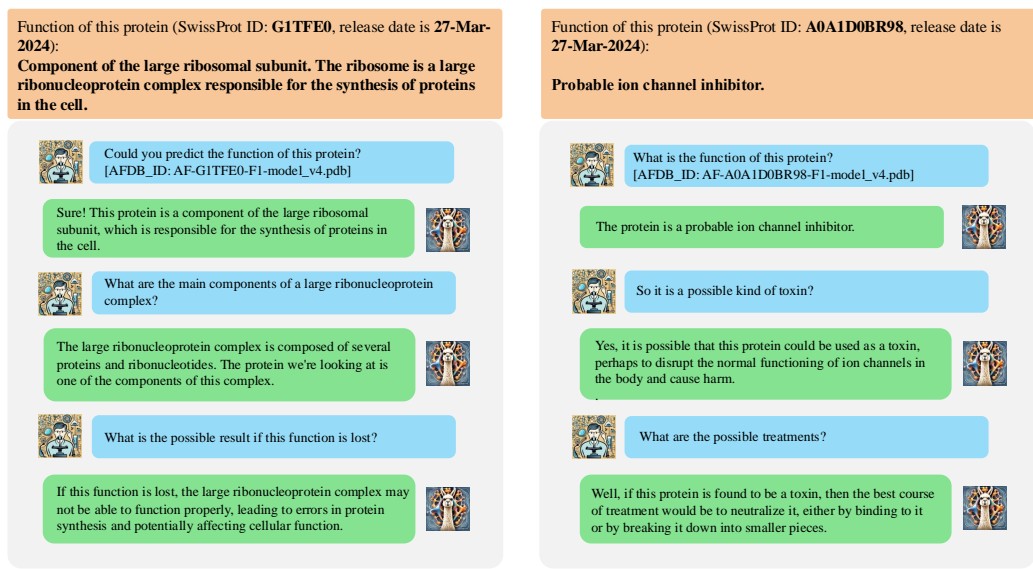

Figure 1: **STELLA's ability to engage in conversation (Model used: STELLA-ESM3-Llama-3.1-8B-Instruct).** The protein G1TFE0 and A0A1D0BR98 are from newly release **2024_02** of the Swiss-Prot database. **Left**: protein G1TFE0. **Right**: protein A0A1D0BR98. **Orange box**: ground truth of the function. **Blue Box**: inquiry from the user. **Green box**: output of the model. Images indicating the user and assistant were generated by AI tools.

## 4 METHODOLOGY

### 4.1 STELLA MODEL ARCHITECTURE

**Overview.** STELLA is a multimodal LLM for protein modeling, drawing inspiration from LLaVA (Liu et al., 2023b), a prominent multimodal architecture designed for vision-language tasks that integrates vision encoders with LLMs. As illustrated in Figure 2, STELLA is composed of three key components: a **protein structure encoder**, a **modality connector**, and a **LLM**. Similar to the typical two-stage training paradigm employed by LLaVA and other multimodal LLMs such as Bunny (He et al., 2024), STELLA adopts a two-stage multimodal instruction tuning (MMIT) approach, which has proven effective in this study. What differs is that STELLA's two stages of training utilize the same datasets, due to the extreme scarcity of protein instruction data. The prompt templates for training are provided in A.2, and hyperparameters in Table 6 (Appendix A.3).

**Protein structure encoder.** The protein structure encoder is responsible for translating protein tertiary structures into high-dimensional structural representations. In this study, we utilize ESM3 (Hayes et al., 2024), a leading model pretrained on multiple modalities, including sequence, structure, and function tokens. ESM3 encodes these distinct modalities as discrete token tracks and integrates them into a unified representation space through transformer blocks. Notably, the model incorporates geometric attention in its initial transformer block, effectively capturing atomic-level details of proteins.

**Modality connector.** The modality connector acts as a bridge between the structural representations derived from the protein structure encoder and the natural language embeddings, such as function descriptions. In this implementation, a simple linear layer is employed as the adapter, which has proven effective, as demonstrated in previous works like LLaVA (Liu et al., 2023b).

**LLM.** The LLM integrated into STELLA is Llama-3.1-8B-Instruct (Dubey et al., 2024), a highly capable model that excels across multiple benchmarks, including general knowledge (Hendrycks et al., 2021a; Wang et al., 2024b; Zhou et al., 2023), mathematics (Cobbe et al., 2021; Hendrycks et al., 2021b; Rein et al., 2023; Clark et al., 2018), code generation (Chen and et al., 2021; Liu et al., 2023c), tool-use (Yan et al., 2024; Srinivasan et al., 2023), long context tasks (Zhang et al., 2024) and multilingual ability (Shi et al., 2022). Additionally, the model exhibits strong safety features, supported by Llama Guard 3, ensuring reliable performance across sensitive applications.

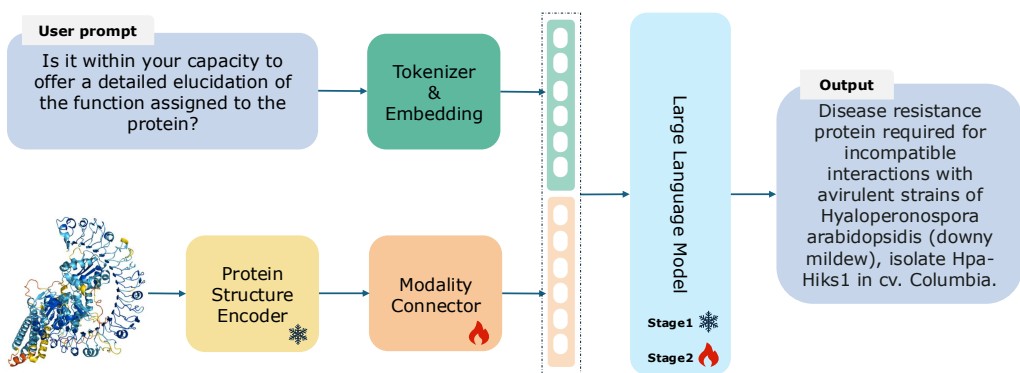

Figure 2: **The architecture of STELLA. Stage1 of MMIT:** to fine-tune the modality connector using the OPI-Struc dataset by freezing the protein structure encoder and LLM. **Stage2 of MMIT:** to continually fine-tune the modality connector and the LLM simultaneously with different learning rates, by freezing the protein structure encoder. **Flame**: model is trainable; **Snowflake**: model is frozen. Protein image credits: AFDB.

## 4.2 TASK DEFINITION

**Functional description prediction (FP).** Through multimodal instruction tuning, STELLA effectively aligns protein structural representations with natural language, enabling the accurate prediction of protein functions from tertiary structures. By leveraging multimodal instruction data, STELLA can uncover novel functional associations, substantially reducing the labor-intensive process of manual annotation. This approach offers a powerful and flexible tool for protein functional description prediction. Furthermore, the integration of LLM-based multi-turn dialogues supports iterative interactions with researchers, facilitating continuous refinement of predictions. This adaptive learning process, driven by expert feedback, not only enhances the model's performance but also allows for tailored adjustments to meet specific research objectives.

**Enzyme-catalyzed reaction prediction (EP).** Predicting enzyme-catalyzed reactions aim at forecasting the biochemical outcomes facilitated by enzymes. Enzymes, as protein-based biological catalysts, are essential for accelerating chemical reactions by lowering activation energy barriers. Accurate prediction of enzyme-catalyzed reactions holds substantial value across various domains, including drug discovery, metabolic engineering, and synthetic biology. In this study, enzyme-catalyzed reactions were mapped to their corresponding enzyme names, which serve as proxies for the reactions in which the associated proteins are involved. This approach allows for more seamless integration with LLMs, ensuring the EP task effectively captures the biological functions of enzymes in a way that aligns with the capabilities of LLMs.

## 4.3 OPI-STRUC DATASET

**Overview.** The **Open Protein Instructions for Structures (OPI-Struc)** dataset was specifically curated to support multimodal instruction tuning (MMIT) in this study, by integrating both protein structural and textual modalities. Corresponding to the **FP** and **EP** task, OPI-Struc is organized into two main categories: **Function** and **Enzyme** (see Appendix A.7, example ④). The **Function** dataset is further divided into two subcategories: $\text{Func}_{ft}$ (see Appendix A.7, example ①) and $\text{Func}_{mc}$ (see Appendix A.7, example ③) based on label formats: free-text question-answer (ft) and multiple-choice question-answer (mc), respectively. Additionally, to reflect the iterative nature of scientific discovery, 20% (49,663 samples) of the $\text{Func}_{ft\_train}$ dataset were randomly selected to be augmented with enriched function annotations generated through conversations using Llama-2-13B-Chat, forming the $\text{Func}_{ft\_train\_aug}$ dataset (see Appendix A.7, example ②). The splitted training and testing sets and corresponding statistics are presented in Table 1.

Table 1: **Statistics of OPI-Struc**. The FP task is composed of two subtasks: $\text{FP}_{ft}$ and $\text{FP}_{mc}$. In the $\text{FP}_{ft}$ task, besides the hold-out testing set $\text{Func}_{ft\_test}$, a newer release of Swiss-Prot v2024_01 (v2401) was utilized to construct $\text{Func}_{ft\_test\_v2401}$ that aims to assess STELLA's performance on unseen data. In the $\text{FP}_{mc}$ task, we designed two versions of testing sets: $\text{Func}_{mc\_test\_1x}$ (options w/o permutation) and $\text{Func}_{mc\_test\_4x}$ (options w/ permutation). See Appendix A.7 for data examples ①, ②, ③ and ④.

| Task | Training set | Training set size | Testing set | Testing set size | Metrics | Protein source |
|---|---|---|---|---|---|---|
| $\text{FP}_{ft}$ | $\text{Func}_{ft\_train}$ (+aug) | 248,315 (+49,663) | $\text{Func}_{ft\_test}$ $\text{Func}_{ft\_test\_v2401}$ | 4,203 270 | BLEU-4 BERT-score ROUGE | AFDB |
| $\text{FP}_{mc}$ | $\text{Func}_{mc\_train}$ | 24,000 | $\text{Func}_{mc\_test\_1x}$ $\text{Func}_{mc\_test\_4x}$ | 4,203 16,812 | Accuracy | AFDB |
| EP | $\text{Enzyme}_{train}$ | 29,205 | $\text{Enzyme}_{test}$ | 5,651 | Accuracy | PDB |

**Data explanation.** Each sample of the OPI-Struc dataset consists of a protein tertiary structure (sourced from either AFDB or PDB), task-specific natural language instructions formatted as conversations, and corresponding labels. In the **Function** dataset, protein structures are derived from AFDB, while the labels (i.e., protein function descriptions) are from the release 2022_04 [3] of Swiss-Prot [4]. In addition, when curating $\text{Func}_{mc\_train}$, the four answer options (A, B, C, D) were randomly permuted within the training set to introduce variability and mitigate answer bias. For the testing set $\text{Func}_{mc\_test}$, two versions were generated: one without permuted answer options ($1x$) and another with permutation ($4x$), ensuring a more robust evaluation by accounting for both consistent and variable answer configurations. The **Enzyme** dataset was obtained from the SIFTS database (Dana et al., 2018), and the original labels, defined by Enzyme Commission (EC) numbers, were mapped to enzyme names using the BRENDA Enzyme Database [5] (e.g., *1.1.1.10 → L-xylulose reductase*). To ensure consistency and accuracy, the OPI-Struc dataset underwent a rigorous preprocessing pipeline following established data cleaning protocols. In addition, detailed analysis of various dataset characteristics were conducted to highlight its comprehensiveness and potential implications for model performance. For instance, the distribution of protein sequence lengths, which correlates with the complexity of protein structures, was examined (see Figure 5, Appendix A.4). These variations underscore the dataset's coverage of a wide range of structural complexities, which is crucial for training models that can generalize effectively across both simple and complex protein structures. Furthermore, the label distribution was analysed, including the length distribution of function descriptions and the frequency of enzyme names, as shown in Figure 6 (Appendix A.4). These insights emphasize the importance of ensuring model robustness across diverse structural and functional complexities to achieve reliable and consistent performance during evaluation.

**Instruction preparation.** The raw data were transformed into an instruction-based format to support learning tasks by providing diverse and structured task instructions. To achieve variation in instruction phrasing, ChatGPT (GPT-3.5) was employed via a web interface to generate rephrased instructions.

---

[3] https://ftp.uniprot.org/pub/databases/uniprot/previous_releases/
release-2022_04/knowledgebase/UniProtKB_SwissProt-relstat.html

[4] https://www.uniprot.org/uniprotkb?query=reviewed:true

[5] https://www.brenda-enzymes.org/

For instance, using the query: *"Could you provide 100 alternative ways to rephrase the prompt 'Please describe the function of the protein'?"*, approximately 100 distinct variations of task instructions were produced (see Appendix A.5 for a detailed list). Each generated instruction was carefully reviewed for accuracy and relevance, ensuring that only high-quality variations were included in the final **Function** dataset. During the augmentation process for the Function-aug$_{train\_FTQA}$ dataset, the Llama-2-13B-Chat model (Touvron et al., 2023) was utilized to generate dialogic interactions based on protein function descriptions sourced from Swiss-Prot. The prompt used for this augmentation was: *"Given a functional description of the protein, design two or three rounds of questions and answers based on this description. Ensure the content is detailed. The output format is: ['Q':, 'A':, 'Q':, 'A':]."* By integrating diverse instructions, this approach facilitated a more dynamic and engaging bridge between protein structural and textual modalities, thereby enriching the OPI-Struc dataset and improving its adaptability and effectiveness for addressing a wide range of research objectives.

**Data split.** (1) The **Function** dataset was divided according to the data split method used in (Abdine et al., 2023), maintaining less than 40% sequence similarity between the protein sequences in the training and testing sets to ensure a rigorous evaluation. (2) The **Enzyme** dataset was partitioned following the same split method as in (Hermosilla et al., 2021).

# 5 EVALUATION OF STELLA MODEL

This study is critical for advancing our understanding of how multimodal LLMs can effectively leverage protein structural representations to address protein-related tasks and extend beyond these applications. By systematically evaluating the STELLA model across the **FP** and **EP** tasks, we seek to elucidate both the strengths and limitations of structural representations in the context of building multimodal LLMs for protein modeling. For these tasks, we designed **five distinct evaluations** based on the corresponding testing sets detailed in Table 1, including $\mathbf{FP}_{ft\_eval}$, $\mathbf{FP}_{ft\_eval\_v2401}$, $\mathbf{FP}_{mc\_eval\_1x}$, $\mathbf{FP}_{mc\_eval\_4x}$, $\mathbf{EP}_{eval}$. The hyperparameters for evaluation are presented in Appendix A.3, while the user prompts for evaluation are listed in Table 7 (Appendix A.6).

Experimental results demonstrate that STELLA is a robust and highly adaptable multimodal LLM. By integrating protein structural representations and LLMs, STELLA exhibits enhanced flexibility and scalability across diverse protein-related tasks, consistently delivering accurate and contextually appropriate outputs. In addition to these strengths, STELLA achieves competitive performance in function and enzyme prediction tasks, rivalling existing specialized models. These results underscore STELLA's potential as a powerful tool for advancing protein science, offering new possibilities for the broader field of computational biology.

## 5.1 EVALUATION METRICS

Multiple typical metrics for natural language processing (NLP) tasks, including BLEU, BERT-score, and ROUGE, were employed for comprehensive evaluation in the **FP** task. However, given the specialization and complexity of biological function descriptions, the quality of LLM responses cannot be fully captured by solely NLP metrics. Recognizing the limitations of such conventional NLP metrics in protein-related tasks, we intentionally designed the **multiple-choice QA (MCQA) subtask, $\mathbf{FP}_{mc}$**, which adopted Accuracy as metrics, to objectively assess STELLA's performance. BLEU, typically applied in machine translation, is used to assess the similarity between two sequences. Particularly, BLEU-4, which measures the overlap of 4-grams between the generated and reference text, was adopted in this study. BERT-score evaluates the token-level similarity between a generated sentence and a reference sentence. ROUGE, a set of metrics traditionally used for automatic text summarization and machine translation, compares generated text to reference texts to calculate the degree of overlap. It includes ROUGE-1, ROUGE-2, and ROUGE-L, which are based on different n-gram methods. ROUGE-L, which focuses on the longest common subsequence, is particularly effective in evaluating summarization and translation quality by considering overall sentence structure. In addition, the **EP** task adopted Accuracy as metrics.

## 5.2 EVALUATION RESULTS

### 5.2.1 RESULTS OF FUNCTION DESCRIPTION PREDICTION

In order to assess STELLA's capability to predict protein functional descriptions based on tertiary structures, we designed the $\mathbf{FP}_{ft\_eval}$, using the hold-out testing set $\mathbf{Func}_{ft\_test}$, which was also utilized for evaluation in Prot2Tex (Abdine et al., 2023). As shown in Table 2, STELLA demonstrated state-of-the-art (SOTA) overall performance, surpassing Prot2Text$_{BASE}$ and Prot2Text$_{LARGE}$ (Abdine et al., 2023) in the $\mathbf{FP}_{ft\_eval}$.

**Comparison with Foldseek.** We adopt Foldseek as baseline comparison, indulging two steps: structure retrieval using Foldseek (Van Kempen et al., 2024) and function mapping from Swiss-Prot. In the first step, for the 4,203 structures in our testing set, we used the Foldseek easy-search [6] command with default parameters to search for similar protein structures within the training set for each test protein. For the e-value parameter, only matches with an e-value below 0.001 are considered and returned. In the second step, the corresponding functional description is determined based on the top-1 retrieved protein from the Swiss-Prot database. The median e-value of the top-1 retrieved proteins is 2.723e-20, indicating a high confidence in the retrieval results by Foldseek.

Table 2: **Evaluation results of the FP task, comparing with existing work.** Training recipes for STELLA-ESM3-Llama-3.1-8B-Instruct: Func$_{ft\_train}$ dataset. **Bold** and underline indicate the best and the runner-up performance.

| Evaluation | Model/Method | BLEU-4 ↑ | BERT Score ↑ | ROUGE Score ↑ | | |
| --- | --- | --- | --- | --- | --- | --- |
| | | | | ROUGE-1 | ROUGE-2 | ROUGE-L |
| | Prot2Text$_{BASE}$ | 0.3511 | 0.8430 | 0.5059 | 0.4271 | 0.4849 |
| | Prot2Text$_{LARGE}$ | 0.3629 | 0.8520 | 0.5368 | 0.4560 | 0.5140 |
| FP$_{ft\_eval}$ | STELLA-ESM3-Llama-3.1-8B-Instruct (e3+e3) | 0.4024 | 0.8496 | 0.5218 | 0.4487 | 0.5041 |
| | STELLA-ESM3-Llama-3.1-8B-Instruct (e3+e6) | **0.4300** | **0.8564** | **0.5423** | **0.4747** | **0.5257** |
| | Foldseek | 0.3627 | 0.8358 | 0.4799 | 0.4027 | 0.4586 |

Furthermore, it is noteworthy that $\mathbf{FP}_{ft\_eval}$ may be impacted by linguistic variability, where model-generated responses with correct meanings differ in expression from the reference. Therefore, we designed $\mathbf{FP}_{mc\_eval\_1x}$ and $\mathbf{FP}_{mc\_eval\_4x}$ to eliminate ambiguity by providing predefined answer choices, which enables more objective and standardized evaluation. This method requires STELLA to not only identify the correct answer but also engage in reasoning and option filtering based on contextual knowledge, thus providing a more comprehensive assessment of its reasoning capabilities. This provides a more robust evaluation for STELLA. Our experiments demonstrated that STELLA exhibits strong reasoning capabilities by achieving accuracies at $\mathbf{Acc@FP}_{mc\_eval\_1x} = \mathbf{0.8056}$ and $\mathbf{Acc@FP}_{mc\_eval\_4x} = \mathbf{0.7618}$. Notably, without integrating with LLMs, baseline models like vanilla ESM3 and Prot2Text are unable to produce outputs in a MCQA format.

### 5.2.2 RESULTS OF ENZYME NAME PREDICTION

$\mathbf{EP}_{eval}$ aims to assess STELLA's ability in enzyme name prediction. On top of the original $\mathbf{Enzyme}_{train}$ set, we excluded 10 samples due to their associated PDB files lacking certain atom coordinates necessary for embedding extraction with the protein structure encoder in STELLA. As shown in Table 3, we witnessed the performance from **Accuracy = 0.8806** to **Accuracy = 0.8885**, by increasing the training epoch from 3 to 6 in stage-2 training. At last, STELLA achieved a state-of-the-art result in the **EP** task, surpassing previous SOTA **Accuracy = 0.8850** in CDConv (Fan et al., 2022).

## 5.3 ABLATION STUDY

### 5.3.1 ABLATION OF PROTEIN ENCODERS AND LLMS

To further investigate the representative ability of different protein encoders, we visualized 4,203 protein structure embeddings from the testing set, Func$_{ft\_test}$, generated by ESM3, Prot2Text (Abdine et al., 2023), and SaProt (Su et al., 2023), using UMAP, as illustrated in Figure 3. The visualization reveals that for the five most frequently occurring functions in the testing set, proteins with the same

---

[6] https://github.com/steineggerlab/foldseek?tab=readme-ov-file#search

Table 3: **Evaluation results of the EP task**. Accuracy is a metric that means the predict answer totally matches the target. **Single:** Enzyme$_{train}$ dataset, **mix3:** Func$_{ft\_train}$ + Func$_{mc\_train}$ + Enzyme$_{train}$. **Bold** and underline indicate the best and the runner-up performance.

| Model | Training manner | Acc@EP ↑ |
|---|---|---|
| DeepFRI (Gligorijević et al., 2021) | w/ pretrain | 0.6330 |
| UniRep (Alley et al., 2019) | w/o pretrain | 0.7290 |
| 3DCNN (Derevyanko et al., 2018) | w/o pretrain | 0.7880 |
| HH-suite3 (Steinegger et al., 2019) | w/o pretrain | 0.8260 |
| ESM-1b (Rives et al., 2021) | w/ pretrain | 0.8310 |
| GearNet-Edge-IEConv (Zhang et al., 2022) | w/o pretrain | 0.8530 |
| IEConv (Hermosilla et al., 2021) | w/o pretrain | 0.8720 |
| GearNet-Multiview-Contrast (Zhang et al., 2022) | w/ pretrain | 0.8750 |
| New IEConv (Hermosilla and Ropinski, 2022) | w/ pretrain | 0.8810 |
| CDConv (Fan et al., 2022) | w/o pretrain | 0.8850 |
| STELLA-ESM3-Llama-3.1-8B-Instruct(single,two-stage,e3+e3) | MMIT | 0.8806 |
| STELLA-ESM3-Llama-3.1-8B-Instruct(single,two-stage,e3+e6) | MMIT | **0.8885** |

function tend to form more compact clusters in the ESM3 representation space compared to the other two encoders. A detailed description of the three encoders is provided in Table 8 (Appendix A.8). Furthermore, several leading LLMs, outlined in Table 9 (Appendix A.9), were integrated into the STELLA framework, enabling an analysis of their impact on STELLA's performance. The ablation results in Table 4 indicate that the combination of the ESM3 encoder with the Llama-3.1 model yielded the best performance in protein function prediction tasks. Moreover, the results underscore the strong overall performance of Llama models across various encoders, reaffirming the effectiveness of combining protein structural information with LLM-based reasoning capabilities.

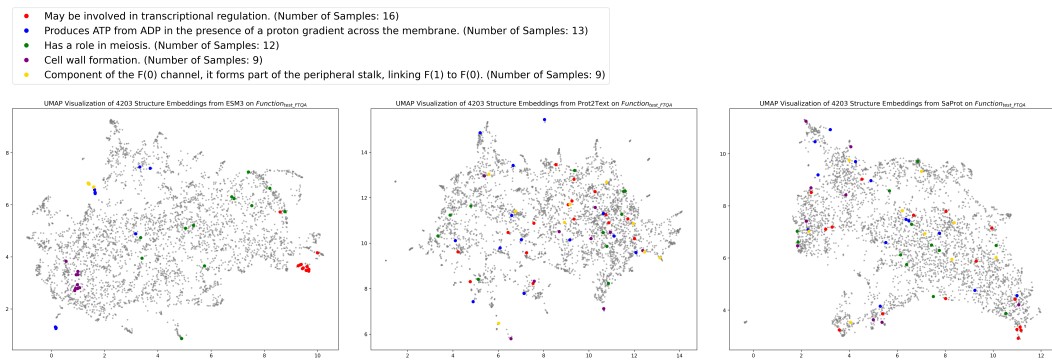

Figure 3: **UMAP visualization of 4,203 protein structure embeddings in the testing set Func**$_{ft\_test}$ **generated by ESM3, Prot2Text, and SaProt.** Each plot illustrates the clustering of protein structures based on their embeddings, revealing the representational differences among the three encoders. The highlighted proteins belong to specific functions as detailed in the legend. ESM3 demonstrates the strongest representative ability.

### 5.3.2 ABLATION OF TRAINING DATA MIX AND TRAINING EPOCHS

An ablation study was conducted to evaluate model performance across varying training data mixes and training epochs. The results, presented in Table 5, indicate that increasing training epochs consistently enhances performance across all data mix configurations. Notably, the model trained exclusively on the **Func**$_{ft\_train}$ dataset achieved the highest evaluation scores when trained for three epochs (e3+e3), suggesting that a longer training duration significantly improves its capability to generate accurate and contextually relevant responses. Incorporating the **Func**$_{mc\_train}$ dataset endowed STELLA with multi-choice Q&A capabilities, while causing only a slight decline in its predictive performance on **FP**$_{ft\_eval}$, as both datasets belong to the same overarching task domain. However, the inclusion of the **Enzyme**$_{train}$ dataset in the mix3 configuration led to superior enzyme prediction performance but caused a noticeable decline in function prediction capability, highlighting the challenges inherent in designing high-quality multitask datasets. Furthermore, during the mix3 training, all metrics demonstrated consistent improvement with extended training, progressing from

Table 4: **Ablation of protein encoders and LLMs in the FP$_{ft\_eval}$.** Training recipes: single Func$_{ft\_train}$ dataset, epochs of two stages (e3+e3). **Bold** and underline indicate the best and the runner-up performance.

| Evaluation | Model | BLEU-4 ↑ | BERT Score ↑ | ROUGE Score ↑ | | |
|---|---|---|---|---|---|---|
| | | | | ROUGE-1 | ROUGE-2 | ROUGE-L |
| FP$_{ft\_eval}$ | STELLA-ESM3-Llama-3.1-8B-Instruct | **0.4024** | 0.8496 | 0.5218 | **0.4487** | **0.5041** |
| | STELLA-ESM3-Llama-3-8B-Instruct | 0.4020 | 0.8503 | 0.5138 | 0.4478 | 0.5001 |
| | STELLA-ESM3-Phi-3-mini-128k-instruct | 0.3807 | 0.8435 | 0.4991 | 0.4273 | 0.4839 |
| | STELLA-Prot2Text-Llama-3.1-8B-Instruct | 0.4009 | 0.8497 | **0.5284** | 0.4454 | 0.5031 |
| | STELLA-Prot2Text-Llama-3-8B-Instruct | 0.3892 | 0.8456 | 0.5177 | 0.4329 | 0.4915 |
| | STELLA-Prot2Text-Phi-3-mini-128k-instruct | 0.3771 | 0.8426 | 0.5058 | 0.4210 | 0.4799 |
| | STELLA-Prot2Text-Mistral-7B-Instruct-v0.2 | 0.3889 | 0.8525 | 0.5224 | 0.4359 | 0.4949 |
| | STELLA-Prot2Text-BioMedGPT-LM-7B | 0.3999 | 0.8488 | 0.5282 | 0.4447 | 0.5020 |
| | STELLA-Prot2Text-BioMistral-7B-DARE | 0.3870 | **0.8533** | 0.5241 | 0.4357 | 0.4980 |
| | STELLA-SaProt-Llama-3-8B-Instruct | 0.3588 | 0.8276 | 0.4685 | 0.3965 | 0.4523 |
| | STELLA-SaProt-Mistral-7B-Instruct-v0.2 | 0.3514 | 0.8251 | 0.4607 | 0.3894 | 0.4455 |
| FP$_{ft\_eval\_v2401}$ | STELLA-ESM3-Llama-3.1-8B-Instruct | 0.0489 | 0.7565 | 0.2210 | **0.1085** | 0.1867 |
| | STELLA-Prot2Text-Llama-3.1-8B-Instruct | 0.0425 | 0.7555 | 0.2454 | 0.1020 | 0.1919 |
| | STELLA-Prot2Text-Llama-3-8B-Instruct | **0.0510** | 0.7605 | 0.2486 | 0.1062 | 0.1918 |
| | STELLA-Prot2Text-Mistral-7B-Instruct-v0.2 | 0.0440 | **0.7685** | **0.2529** | 0.1046 | **0.1975** |

(e3+e1) to (e3+e3), as illustrated in Figure 7 (AppendixA.10). This trend underscores the positive effect of prolonged training on model performance and emphasizes the significance of meticulous dataset selection and appropriate training duration to optimize predictive performance. Additionally, the **FP**$_{ft\_eval\_v2401}$ was designed to assess STELLA's generalization capability on newly released proteins, using the testing set **Func**$_{ft\_test\_v2401}$.

Table 5: **Ablation of training data mix and training epochs across FP$_{ft\_eval}$, FP$_{mc\_eval\_1x}$, FP$_{mc\_eval\_4x}$ and EP$_{eval}$ for STELLA-ESM3-Llama-3.1-8B-Instruct. single:** Func$_{ft\_train}$, **mix2:** Func$_{ft\_train}$ + Func$_{mc\_train}$, **mix3:** Func$_{ft\_train}$ + Func$_{mc\_train}$ + Enzyme$_{train}$. The 2nd column indicates the training epochs of two stages. **Bold** indicates the best performance in each configuration.

| Data mix | Training epochs | BLEU-4 ↑ | BERT Score ↑ | ROUGE Score ↑ | | | Acc@FP$_{mc\_eval}$ ↑ | | Acc@EP$_{eval}$ ↑ |
|---|---|---|---|---|---|---|---|---|---|
| | | | | ROUGE-1 | ROUGE-2 | ROUGE-L | 1x | 4x | |
| single | (e3+e1) | 0.2653 | 0.8065 | 0.3938 | 0.3097 | 0.3770 | - | - | - |
| | (e3+e2) | 0.3574 | 0.8363 | 0.4790 | 0.4028 | 0.4617 | - | - | - |
| | (e3+e3) | **0.4024** | **0.8496** | **0.5218** | **0.4487** | **0.5041** | - | - | - |
| mix2 | (e3+e1) | 0.2397 | 0.8003 | 0.3624 | 0.2861 | 0.3505 | 0.6936 | 0.5893 | - |
| | (e3+e2) | 0.3411 | 0.8330 | 0.4554 | 0.3878 | 0.4428 | 0.7940 | 0.7428 | - |
| | (e3+e3) | 0.4020 | 0.8491 | 0.5119 | 0.4465 | 0.4980 | **0.8056** | **0.7618** | - |
| mix3 | (e3+e1) | 0.1092 | 0.7665 | 0.1749 | 0.1352 | 0.1747 | 0.7345 | 0.6460 | 0.7972 |
| | (e3+e2) | 0.1948 | 0.7898 | 0.2754 | 0.2254 | 0.2687 | 0.7904 | 0.7307 | 0.8666 |
| | (e3+e3) | 0.2394 | 0.8025 | 0.3233 | 0.2720 | 0.3151 | 0.7956 | 0.7402 | **0.8809** |

# 6 CONCLUSION AND FUTURE WORK

This study presented STELLA, a novel multimodal LLM in an early fusion strategy inherited from ESM3, which integrates protein structural representations with the contextual knowledge embedded within LLMs. Through multimodal instruction tuning using the OPI-Struc dataset, STELLA achieves SOTA performance in two critical tasks: protein functional description prediction and enzyme-catalyzed reaction prediction. Beyond excelling in protein understanding, STELLA establishes an innovative LLM-based paradigm for protein-related research, complementing traditional pLM-based approaches. This work highlights the transformative potential of STELLA as a powerful computational tool for advancing protein biology. Looking ahead, future efforts should focus on encompassing more downstream tasks by expanding the OPI-Struc dataset, as well as exploring advanced techniques such as retrieval-augmented generation (RAG) and agent-based systems to further enhance STELLA's capabilities. These advancements will unlock its full potential as a pioneering tool in computational biology, solidifying its role in driving the next generation of innovations in protein science and beyond.

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

# A APPENDIX

## A.1 EXAMPLE DEMONSTRATION OF STELLA'S CAPABILITIES THROUGH CASE STUDIES

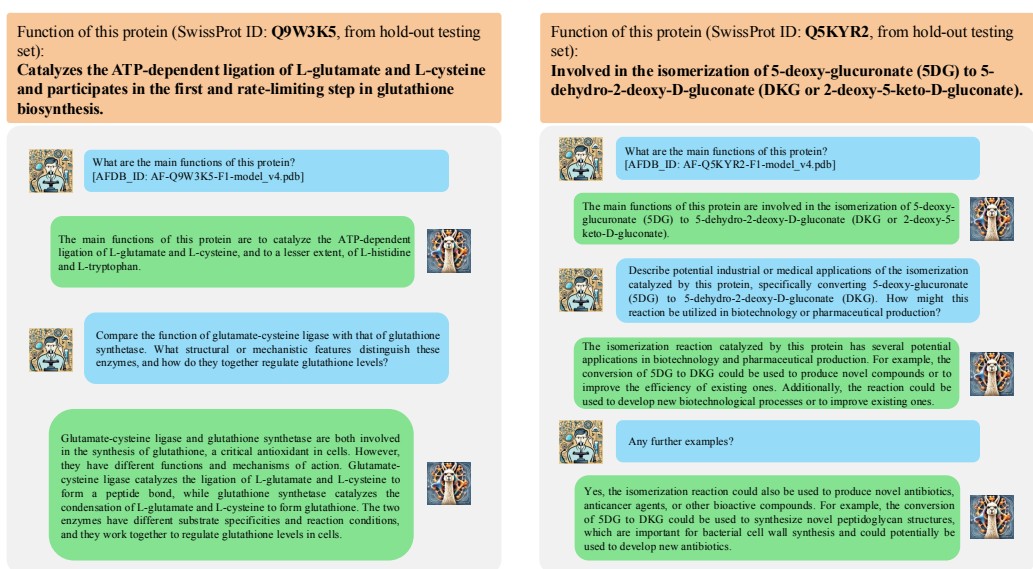

Figure 4: **STELLA's ability to engage in conversation (Model used: STELLA-ESM3-Llama-3.1-8B-Instruct).** The protein Q9W3K5 and Q5KYR2 are from the hold-out testing set of OPI-Struc. **Left**: protein Q9W3K5. **Right**: protein Q5KYR2. **Orange box**: ground truth of the function. **Blue Box**: inquiry from the user. **Green box**: output of the model. Images indicating the user and assistant were generated by AI tools.

## A.2 PROMPT TEMPLATE FOR TRAINING

> **The prompt template of STELLA-Prot2Text-Llama-3.1-8B-Instruct**
>
> <|begin_of_text|><|start_header_id|>user<|end_header_id|>
>
> <structure>
> May I request a comprehensive breakdown outlining the function linked to the protein?
> <|eot_id|><|start_header_id|>assistant<|end_header_id|>
> Involved in the gluconeogenesis. Catalyzes stereospecifically the conversion of dihydroxyacetone phosphate (DHAP) to D-glyceraldehyde-3-phosphate (G3P). <|eot_id|><|end_of_text|>

> **The prompt template of STELLA-Prot2Text-Mistral-7B-Instruct-v0.2**
>
> [INST] <structure>
> May I request a comprehensive breakdown outlining the function linked to the protein? [/INST]Involved in the gluconeogenesis. Catalyzes stereospecifically the conversion of dihydroxyacetone phosphate (DHAP) to D-glyceraldehyde-3-phosphate (G3P)

## A.3 HYPERPARAMETERS FOR TRAINING AND EVALUATION

Stage1 aims to align a protein structure embedding space and a plain-text embedding space. In this stage, the modality connector trainable, while both the protein structure encoder and the LLM are frozen. Stage2 is dedicated to teach STELLA to follow complicated natural language instructions and generate response dedicated to protein tasks. In this stage, both the modality connector and the LLM are trainable with different learning rates, while the protein structure encoder is still frozen.

Both stages use the same training datasets. The prompts templates for training follow the examples shown in Appendix A.2.

Hyperparameters in PT stage and IT stage are summarized in Table 6. It is noteworthy that we adopt different learning rates for each different components of STELLA to finely control the training process. Especially, in the IT stage, we set the learning rate of the modality connector larger than LLM backbone, to improve LLMs' training convergence.

Table 6: **Hyperparameters for stage1 training, stage2 training and testing.** FFT: Full Fine-tuning; LoRA: LoRA Tuning

| Config | Stage1 | Stage2 | Testing |
|---|---|---|---|
| DeepSpeed ZeRO Stage | 2 | 3 | NA |
| optimizer | AdamW | AdamW | NA |
| optimizer hyperparameters | $(\beta_1,\beta_2)$=(0.9, 0.999), eps=1e-8 | $(\beta_1,\beta_2)$=(0.9, 0.999), eps=1e-8 | NA |
| per_device_train_batch_size | 2 | 1(FFT)/2(LoRA) | NA |
| gradient_accumulation_steps | 4 | 2(FFT)/4(LoRA) | NA |
| gradient_checkpointing | True | True | NA |
| learning rate (lr) | 2e-5 (Connector) | 2e-4 (Connector), 2e-5 (LLM) | NA |
| weight decay | 0.0 | 0.0 | NA |
| warmup steps | 48 | - | NA |
| warmup ratio | - | 0.03 | NA |
| lr scheduler type | cosine | cosine | NA |
| training epochs | 3 | 3 | NA |
| GPU | 4*A100 | 8*A100(FFT)/4*A100(LoRA) | 1*A100 |
| temperature | NA | NA | 0.2 |
| top_k | NA | NA | 50 |
| top_p | NA | NA | 0.75 |
| num_beams | NA | NA | 1 |
| max_new_tokens | NA | NA | 1000 |
| use_cache | NA | NA | True |
| do_sample | NA | NA | True |

## A.4 ANALYSIS OF DATA LABEL DISTRIBUTION OF THE OPI-STRUC DATASET

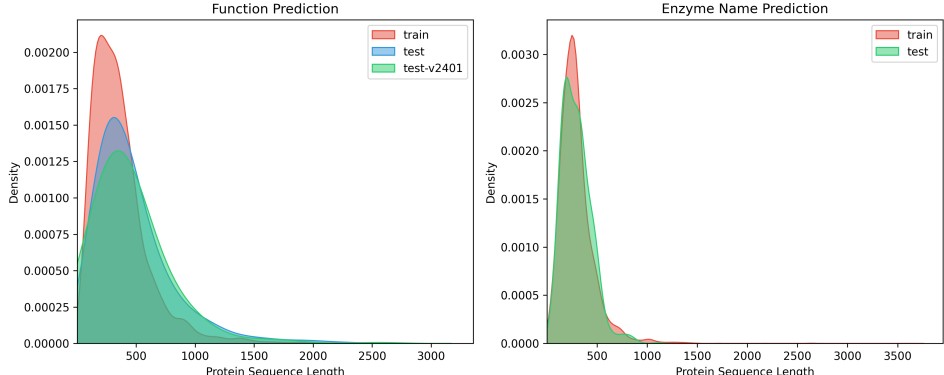

Figure 5: **Distribution of protein sequence lengths across the FP (left) and EP (right) tasks for training and testing sets.** The variation in sequence length distribution between the training and testing sets ensures model robustness across proteins with diverse structural complexities.

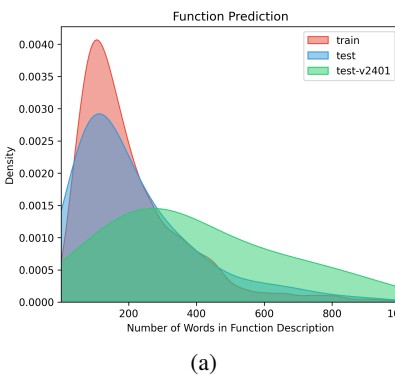 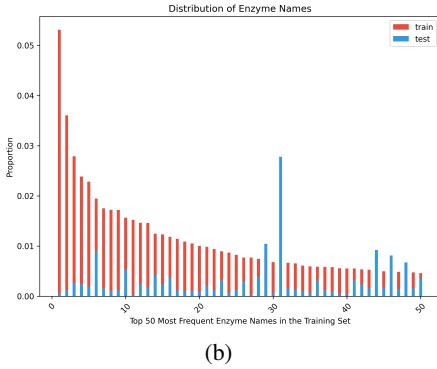

(a)     (b)

Figure 6: **a**: Length distribution of functional descriptions in the Function dataset. **b**: Frequency of enzyme names in the Enzyme dataset. The enzyme name distribution in the training set follows a long-tailed pattern, but the label distribution in the test set differs significantly from that in the training set.

## A.5 EXPANDED INSTRUCTIONS BY CHATGPT (GPT-3.5)

---

**Expanded instructions by ChatGPT (GPT-3.5)**

- May I request an elaborate overview of the function linked to the protein?
- Is it within your capacity to provide a comprehensive overview of the function associated with the protein?
- Can you supply a detailed breakdown of the function ascribed to the protein?
- May I request a comprehensive depiction of the function pertaining to the protein?
- May I request a comprehensive account outlining the function of the protein?
- Is it possible for you to furnish a comprehensive breakdown of the function associated with the protein?
- May I request a comprehensive breakdown outlining the function linked to the protein?'
- Could you share a detailed elucidation of the function assigned to the protein?'
- Would you mind giving me a detailed breakdown of the function associated with the protein?
- Is it within your capacity to provide a comprehensive overview of the function linked to the protein?
- Could you supply an extensive description of the function ascribed to the protein?
- Can you furnish a comprehensive elucidation of the function ascribed to the protein?
- Is it feasible for you to offer a comprehensive analysis regarding the function of the protein?
- Would it be possible for you to offer a thorough breakdown of the function ascribed to the protein?
- Can you furnish a comprehensive explanation regarding the function of the protein?
- Can you furnish a comprehensive analysis of the function encompassing the protein?
- May I inquire about a comprehensive explanation encompassing the function of the protein?
- Can you furnish a comprehensive description of the function ascribed to the protein?
- Would you mind providing a comprehensive overview of the function attributed to the protein?
- Could you share an elaborate overview of the function linked to the protein?
- Could you share a comprehensive overview of the function encompassing the protein?

---

- Could you offer a comprehensive elucidation of the function assigned to the protein?
- May I request a comprehensive breakdown outlining the function associated with the protein?
- Would you mind giving me a comprehensive analysis of the function attributed to the protein?
- Is it within your capacity to offer a detailed elucidation of the function assigned to the protein?
- Can you supply a comprehensive explanation of the function related to the protein?
- Can you give me a comprehensive explanation of the function ascribed to the protein?
- Is it possible for you to provide a detailed description of the function ascribed to the protein?
- Could you share a comprehensive description of the function encompassing the protein?
- Would you mind providing a thorough explanation of the function related to the protein?
- Can you offer a comprehensive analysis of the function attributed to the protein?
- Can you supply a comprehensive depiction of the function related to the protein?
- May I request a detailed overview of the function associated with the protein?
- May I request a comprehensive analysis of the function attributed to the protein?
- Would you mind giving me a comprehensive description of the function attributed to the protein?
- Is it feasible for you to offer a comprehensive explanation regarding the function of the protein?
- Is it within your capacity to provide a comprehensive explanation of the function related to the protein?
- Would it be possible for you to provide a comprehensive analysis of the function attributed to the protein?
- May I inquire about a thorough account of the function related to the protein?
- May I request a comprehensive account of the function pertaining to the protein?
- Is it feasible for you to give an extensive overview of the function linked to the protein?
- Could you provide a detailed elucidation of the function encompassing the protein?
- Would it be possible for you to offer a comprehensive depiction encompassing the function of the protein?
- Is it feasible for you to offer a comprehensive account of the function ascribed to the protein?
- Is it within your capacity to provide a comprehensive breakdown of the function linked to the protein?
- Could you share a comprehensive breakdown of the function linked to the protein?
- May I inquire about a comprehensive depiction of the function encompassing the protein?
- Is it within your capacity to provide a comprehensive overview of the function assigned to the protein?
- May I inquire about a comprehensive account of the function associated with the protein?
- Could you provide a detailed account of the function assigned to the protein?
- Could you furnish a detailed depiction of the function encompassing the protein?
- Can you provide a detailed description of the function ascribed to the protein?
- May I inquire about a comprehensive explanation outlining the function of the protein?
- May I request a comprehensive overview of the function ascribed to the protein?
- Could you provide a detailed elucidation outlining the function associated with the protein?

- Can you provide a comprehensive elucidation of the function assigned to the protein?
- Would it be possible for you to offer a comprehensive explanation of the function associated with the protein?
- Would you mind giving me a comprehensive account of the function attributed to the protein?
- May I inquire about a comprehensive breakdown of the function assigned to the protein?
- Can you give me a detailed breakdown of the function linked to the protein?
- Can you give me a detailed depiction of the function encompassing the protein?
- Is it possible for you to furnish a comprehensive depiction of the function encompassing the protein?
- Can you supply a comprehensive breakdown of the function associated with the protein?
- Can you furnish a detailed overview of the function linked to the protein?
- May I inquire about a thorough explanation of the function related to the protein?
- Could you share a detailed analysis of the function attributed to the protein?
- Would you be able to furnish a detailed explanation of the function encompassing the protein?
- Is it feasible for you to provide an elaborate account of the function attributed to the protein?
- May I inquire about a comprehensive analysis of the function assigned to the protein?
- Would you be able to provide a detailed elucidation of the function assigned to the protein?
- May I request a detailed breakdown of the function associated with the protein?
- Would it be possible for you to offer a comprehensive depiction of the function ascribed to the protein?
- May I inquire about a detailed account of the function assigned to the protein?
- Could you provide an in-depth explanation of the function associated with the protein?
- May I inquire about a detailed description of the function ascribed to the protein?
- Would you be able to provide a comprehensive account of the function pertaining to the protein?
- Can you furnish a comprehensive description outlining the function associated with the protein?
- Can you supply a comprehensive analysis of the function linked to the protein?
- Would it be possible for you to offer a comprehensive analysis of the function related to the protein?
- Could you offer a comprehensive breakdown of the function associated with the protein?
- Could you supply a thorough explanation of the function related to the protein?
- Is it feasible for you to supply a thorough explanation of the function related to the protein?
- Would it be possible for you to offer an in-depth description of the function of the protein?
- Is it within your capacity to provide a comprehensive depiction of the function related to the protein?
- Could you provide a detailed description outlining the function of the protein?
- Can you share a comprehensive account of the function pertaining to the protein?
- Would it be possible for you to provide an extensive description of the function ascribed to the protein?
- Could you share a comprehensive depiction of the function pertaining to the protein?
- Could you provide a detailed analysis of the function ascribed to the protein?
- Is it within your capacity to provide a comprehensive elucidation of the function associated with the protein?

- Would you mind giving me a comprehensive depiction of the function pertaining to the protein?
- Could you share a comprehensive overview of the function ascribed to the protein?
- Is it within your capability to offer a detailed account of the function pertaining to the protein?
- Can you supply a comprehensive account of the function linked to the protein?
- Could you share a comprehensive breakdown of the function ascribed to the protein?
- Would it be possible for you to offer a comprehensive account linked to the function of the protein?
- Can you supply a comprehensive explanation of the function assigned to the protein?
- Is it possible for you to provide a comprehensive analysis of the function attributed to the protein?
- Is it feasible for you to offer a comprehensive description of the function attributed to the protein?

## A.6 PROMPT TEMPLATE FOR EVALUATION

Table 7 presents the user prompts used in the evaluation of three tasks. Notably, we designed the prompt to ensure that the model outputs only one of the four options (A, B, C, or D) in the $FP_{MCQA}$ task, facilitating assessment.

Table 7: **User prompts for evaluation**.

| Task | Testing set | Answer formatting prompts |
|------|-------------|---------------------------|
| $FP_{ft}$ | $Func_{ft\_test}$ $Func_{ft\_test\_v2401}$ | What are the main functions of this protein? |
| $FP_{mc}$ | $Func_{mc\_text\_1x}$ $Func_{mc\_text\_4x}$ | Answer with the option's letter from the given choices directly. Please respond to the question with an answer choice, which is either A, B, C or D. |
| EP | $Enzyme_{test}$ | What is the enzyme name linked to this protein? |

## A.7 EXAMPLES OF THE OPI-STRUC DATA

① An example of $Func_{ft\_train}$ data

```
[
  {
    "swissprot_id": "Q0BWM9",
    "sequence": "
      MFNKQSVSLEWAGRTLTIETGQVARQADGAVMVQYGDTIVLATAVFAKEAKPGQDFFPLTV
      NYQEKYFASGRIPGGFFKREGRPTEKETLTSRLIDRPIRPLFVDGFKHEVQVVVTTLSYDL
      ENDADIIALVGASAALVLSGAPFMGPIGAARVGYKDGQYIINPTIAELEESELDLVVAGTT
      DAVMMVESQAAELSEDVMLGAVVAGHDAMQPVIDAIIALAEKAAKEPFAYEPPDHSAALKS
      VIDTVGADLSKAYKITAKGERYAAIGAAKDKAKAALLGTDEAPGVMTPEVFKTVFKEAEAS
      VVRGDILKTGQRIDGRKLDQIRPIVAEAGFLPRTHGSSLFTRGETQAICVATLGTSDDEQY
      IDGLDGTKKEKFMLHYNFPPYSVGETGRMGGAGRREIGHGKLAWRALKAVLPKHEDFPYTI
      RMVSEITESNGSSSMATVCGCSLAMMDAGVPLTRPVSGIAMGLILEGSEFAVLSDILGDED
      HLGDMDFKVAGTENGVTSLQMDIKVAGITKDIMGKALEQAKGGRMHILGEMGKALTASRGQ
      LSENAPQMEIIKVPTDKIRDVIGSGGKVIRGIVDETGAKVNIDDDGTVQISAMDRKSIDAA
      IKMIKGITAEAEVGEIYEGKVVSMKDFGIFVNFFGPKDGLVHVSQMANKRIGHPKEMVKEG
      DKVWVKLMGFDERGKVRLSMKVVDQETGKELAEEAGDDASED",
    "conversations": [
      {
        "from": "human",
```

```
1080
1081        "value": "Can you furnish a comprehensive description
1082            outlining the function associated with the protein?\n<
1083            structure>"
1084      },
1085      {
1086        "from": "gpt",
1087        "value": "Involved in mRNA degradation. Catalyzes the
1088            phosphorolysis of single-stranded polyribonucleotides
1089            processively in the 3'- to 5'-direction."
1090      }
1091    ],
1092    "AFDB_id": "AF-Q0BWM9-F1-model_v4"
1093  },
1094 ]
```

② An example of Func$_{ft\_train\_aug}$ dataset

```
[
  {
    "swissprot_id": "Q1WT60",
    "sequence": "
      MENKKLHIALLFGGNSSEHDVSKRSAHNIYDALDKDKYDVSVFMFTKKGFLLGNKDSMRIF
      DGENEDDVVTEVIKDVDFSNPLANIQNLAEVKDVDVFYPVIHGNMGEDGTVQGLFRLLNKPW
      IGSGVASSGVSFDKDLTKKLLTLNGIRNTKYVLVTPENKADYPYAKVAEELGETLFVKPARQ
      GSSVGIHKVRNEEEYNAALEDGFKYDYKILVEEAIKNPREVECSVLGNRDIKASKLGAIRIP
      ESDDFYDYNNKFVDASGVVFEMPIKLPEKLTKEIQQMSLDAFRALDNRGLARMDFLVDKNDV
      PYFGEVNTLPGFTNISLYPQLWEVSGISYSELIDQLIQLAIDEFNDNAKIHYDFTKLGTEKV
      GKKIIGE",
    "conversations": [
      {
        "from": "human",
        "value": "What is the main function of this protein?\n<
            structure>"
      },
      {
        "from": "gpt",
        "value": "The main function of this protein is cell wall
            formation."
      },
      {
        "from": "human",
        "value": "Can you explain the process of cell wall formation
            ?"
      },
      {
        "from": "gpt",
        "value": "Certainly! The protein helps to synthesize and
            modify the cell wall, which is a critical structure that
            provides support, protection, and cell-to-cell interaction
            . The cell wall is composed of various components,
            including polysaccharides, proteins, and other molecules.
            Our protein plays a key role in the synthesis and
            modification of these components to form a strong and
            stable cell wall."
      },
      {
        "from": "human",
        "value": "How does the protein contribute to the synthesis of
             the cell wall?"
      },
```

```
      {
        "from": "gpt",
        "value": "The protein helps to catalyze the formation of
            glycosidic bonds between the cell wall components. These
            bonds are crucial for the stability and integrity of the
            cell wall. Additionally, the protein can also modify
            existing glycosidic bonds to further strengthen the cell
            wall."
      }
    ],
    "AFDB_id": "AF-Q1WT60-F1-model_v4"
  },
]
```

③ An example of Func$_{mc\_train}$ data

```
[
  {
    "swissprot_id": "P62877",
    "sequence": "
      MAAAMDVDTPSGTNSGAGKKRFEVKKWNAVALWAWDIVVDNCAICRNHIMDLCIECQANQ
    ASATSEECTVAWGVCNHAFHFHCISRWLKTRQVCPLDNREWEFQKYGH",
    "conversations": [
      {
        "from": "human",
        "value": "<structure>\nWhat are the main functions of this
            protein?\nA. E3 ubiquitin ligase component of multiple
            cullin-RING-based E3 ubiquitin-protein ligase (CRLs)
            complexes which mediate the ubiquitination and subsequent
            proteasomal degradation of target proteins, including
            proteins involved in cell cycle progression, signal
            transduction, transcription and transcription-coupled
            nucleotide excision repair. CRLs complexes and ARIH1
            collaborate in tandem to mediate ubiquitination of target
            proteins, ARIH1 mediating addition of the first ubiquitin
            on CRLs targets. The functional specificity of the E3
            ubiquitin-protein ligase complexes depends on the variable
             substrate recognition components. As a component of the
            CSA complex promotes the ubiquitination of ERCC6 resulting
             in proteasomal degradation. Recruits the E2 ubiquitin-
            conjugating enzyme CDC34 to the complex and brings it into
             close proximity to the substrate. Probably also
            stimulates CDC34 autoubiquitination. May be required for
            histone H3 and histone H4 ubiquitination in response to
            ultraviolet and for subsequent DNA repair. Promotes the
            neddylation of CUL1, CUL2, CUL4 and CUL4 via its
            interaction with UBE2M. Involved in the ubiquitination of
            KEAP1, ENC1 and KLHL41. In concert with ATF2 and CUL3,
            promotes degradation of KAT5 thereby attenuating its
            ability to acetylate and activate ATM.\nB. Part of the
            MIS12 complex which is required for normal chromosome
            alignment and segregation and kinetochore formation during
             mitosis.\nC. Catalyzes the cyanide-resistant oxidation of
             ubiquinol and the reduction of molecular oxygen to water,
             but does not translocate protons and consequently is not
            linked to oxidative phosphorylation. May increase
            respiration when the cytochrome respiratory pathway is
            restricted, or in response to low temperatures.\nD. Pair-
            rule protein expressed in a segmentally repeating pattern
            to define the polarity of embryonic segments. Capable of
```

```
          sequence-specific DNA-binding.\nAnswer with the option's
          letter from the given choices directly."
    },
    {
      "from": "gpt",
      "value": "A"
    }
  ],
  "AFDB_id": "AF-P62877-F1-model_v4"
},
]
```

④ An example of Enzyme$_{train}$ data

```
[
  {
    "PDB_id": "5ivp.A",
    "conversations": [
      {
        "from": "human",
        "value": "Could you inform me of the enzyme name tagged to
            this protein?<structure>\n"
      },
      {
        "from": "gpt",
        "value": "aminoacyl-tRNA hydrolase"
      }
    ]
  },
]
```

## A.8 COMPARISON OF PROTEIN STRUCTURE ENCODERS

Difference among three representative protein structure encoders employed in this study, ESM3, Prot2Text and SaProt, are presented in Table 8.

## A.9 DIFFERENT COMPOSITION OF PROTEIN STRUCTURE ENCODERS AND LLMS

According to the architecture of STELLA, it is flexible and customizable to integrate various protein encoders and LLMs to form STELLA variants. In order to delve into the effectiveness of different composition of protein encoders and LLMs, we elaborately choose different protein encoders and foundation LLMs, as shown in Table 9.

## A.10 ABLATION OF TRAINING EPOCHS FOR MIX3 TRAINING

Each graph in Figure 7 shows how the scores for BLEU-4, BERT Score, ROUGE Scores, and Accuracy change over the training periods labeled as (e3+e1), (e3+e2), and (e3+e3). All the metrics improve as training epochs increase, suggesting better performance with more training.

Table 8: **Comparison of three representative protein structure encoders.**

| Protein encoder | Modality | Modality fusion methods |
|---|---|---|
| **ESM3** | Sequence, Structure, Function | ESM3 is a multimodal model pretrained on massive sequence, structure and function tokens via masked language modeling (MLM). It encodes these modalities as discrete token tracks, which are fused into a unified representation space using several transformer blocks, with geometric attention in the first block to incorporate atomic information. |
| **Prot2Text** | Sequence, Structure, Function | Prot2Text is a multimodal model incorporating a Relational Graph Convolution Network (RGCN), ESM-2 and GPT-2 to generate protein function annotation. It is designed to integrate information from two sources: the output of the RGCN and the protein sequence data processed by ESM-2. The RGCN receives all-atom protein structures as its input, providing detailed structural information. Subsequently, the Prot2Text encoder aligns this integrated data with functional annotation through a generative alignment approach using a text decoder. Prot2Text serve as a method for protein structure-text feature alignment. |
| **SaProt** | Sequence, Structure | SaProt is a large-scale pre-trained model using about 40 million protein sequences and structures with structure-aware vocabulary which integrates residue tokens with structure tokens simultaneously. It adopts an ESM-based architecture that takes inputs as structure-aware protein sequences, which combine the protein sequence residue tokens and discrete structural tokens encoded using folkseek. This encoder is not aligned with functional annotation text. |

Table 9: **Specifications of STELLA composition of various protein structure encoders and foundation LLMs.**

| Protein encoder | Foundation LLM | Note | Composed STELLA variant |
|---|---|---|---|
| ESM3 (Hayes et al., 2024) | Llama-3.1-8B-Instruct (AI@Meta, 2024) | Open source model by Meta | STELLA-ESM3-Llama-3.1-8B-Instruct |
| | Llama-3-8B-Instruct (AI@Meta, 2024) | Open source model by Meta | STELLA-ESM3-Llama-3-8B-Instruct |
| | Mistral-7B-Instruct-v0.2 (Jiang et al., 2023) | Open source model by Mistral AI | STELLA-ESM3-Mistral-7B-Instruct-v0.2 |
| | Phi-3-mini-128k-instruct (Abdin et al., 2024) | Open source model by Microsoft | STELLA-ESM3-Phi-3-mini-128k-instruct |
| | BioMistral-7B-DARE [a] | Tailored model for biomedical domain | STELLA-ESM3-BioMistral-7B-DARE |
| | BioMedGPT-LM-7B [b] Luo et al. (2023) | Tailored model for biomedical domain | STELLA-ESM3-BioMedGPT-LM-7B |
| Prot2Text (Abdine et al., 2023) | Llama-3.1-8B-Instruct | Open source model by Meta | STELLA-Prot2Text-Llama-3.1-8B-Instruct |
| | Llama-3-8B-Instruct | Open source model by Meta | STELLA-Prot2Text-Llama-3-8B-Instruct |
| | Mistral-7B-Instruct-v0.2 | Open source model by Mistral AI | STELLA-Prot2Text-Mistral-7B-Instruct-v0.2 |
| | Phi-3-mini-128k-instruct | Open source model by Microsoft | STELLA-Prot2Text-Phi-3-mini-128k-instruct |
| | BioMistral-7B-DARE | Tailored model for biomedical domain | STELLA-Prot2Text-BioMistral-7B-DARE |
| | BioMedGPT-LM-7B | Tailored model for biomedical domain | STELLA-Prot2Text-BioMedGPT-LM-7B |
| SaProt (Su et al., 2023) | Llama-3.1-8B-Instruct | Open source model by Meta | STELLA-SaProt-Llama-3.1-8B-Instruct |
| | Llama-3-8B-Instruct | Open source model by Meta | STELLA-SaProt-Llama-3-8B-Instruct |
| | Mistral-7B-Instruct-v0.2 | Open source model by Mistral AI | STELLA-SaProt-Mistral-7B-Instruct-v0.2 |
| | Phi-3-mini-128k-instruct | Open source model by Microsoft | STELLA-SaProt-Phi-3-mini-128k-instruct |
| | BioMistral-7B-DARE | Tailored model for biomedical domain | STELLA-SaProt-BioMistral-7B-DARE |
| | BioMedGPT-LM-7B | Tailored model for biomedical domain | STELLA-SaProt-BioMedGPT-LM-7B |

[a] Merge (Yu et al., 2024) of Mistral-7B-Instruct-v0.1 and BioMistral-7B (Labrak et al., 2024) which was further pre-trained on top of Mistral-7B-Instruct-v0.1 using PubMed Central Open Access from https://www.ncbi.nlm.nih.gov/pmc/tools/openftlist/
[b] Increamtally pre-training from Llama-2-7B-Chat with S2ORC (Lo et al., 2020) corpus.

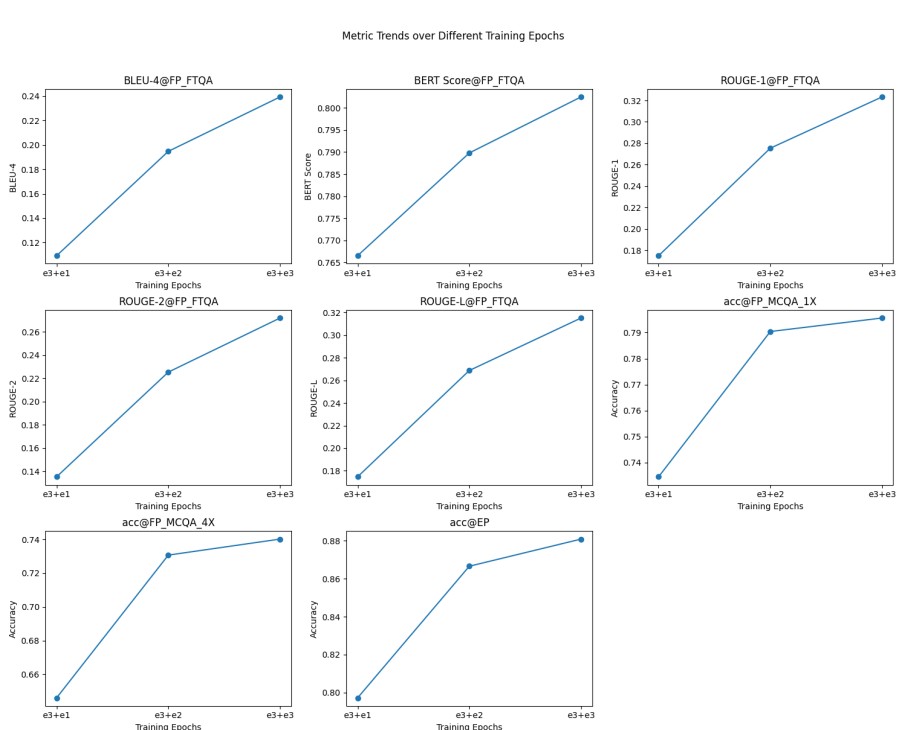

Figure 7: **The trend lines for the various metrics across different training epochs.**

