# OpenReview forum: "STELLA: Leveraging Structural Representations to Enhance Protein Understanding with Multimodal LLMs"
_ICLR.cc/2025/Conference — Submitted to ICLR 2025_

### Official Review · Reviewer_qSap · 2024-10-30

**Soundness:** 3
**Presentation:** 4
**Contribution:** 2
**Rating:** 6
**Confidence:** 3

**Summary:**

The paper introduces STELLA for protein function and enzyme-catalyzed reaction prediction. The work combined multiple existing tools to generate the workflow. While the idea is in principle sound, the novelty is not well described.

**Strengths:**

STELLA integrates protein structure data with LLMs to enhance protein function and enzyme prediction tasks.
It provides comprehensive evaluations, utilizing different datasets and metrics, which adds credibility to the performance claims.
By sharing the code, datasets, and pre-trained models, the study facilitates collaboration and fosters further research in the field.
The presentation, including graph demonstration and writtent text, is clear.

**Weaknesses:**

The paper does not clearly differentiate STELLA from existing multimodal models like Prot2Text and other protein prediction frameworks. A clearer outline of unique contributions and improvements over prior methods would strengthen the work.
The benchmark results are not superior to state-of-the-art results from existing multimodal models.
The paper may include more demonstrations from biological side.

**Questions:**

While STELLA seems to enhance function prediction, I wonder if the model’s reasoning behind those predictions is easy to interpret. What measures, if any, have been taken to make its outputs understandable, especially for biologists who need to validate its findings?
Given that STELLA relies heavily on structured protein data, how does it perform when dealing with less common protein structures? In my experience, we often work with proteins that lack precise structure or even sequence information in certain regions. How well could STELLA handle incomplete protein data?
Does STELLA show a significant enough improvement over other established models to justify its scientific meaning and contribution?
I’m curious about the model’s scalability because of the interactive demonstration. Can STELLA efficiently handle large-scale datasets or high-throughput predictions?

---

> ### Author Response · Authors · 2024-11-25
> **Response to Reviewer qSap -- #1**
>
> Thank you for raising several insightful questions for our work, which are addressed below.
>
> **Questions**
> 1. While STELLA seems to enhance function prediction, I wonder if the model’s reasoning behind those predictions is easy to interpret. What measures, if any, have been taken to make its outputs understandable, especially for biologists who need to validate its findings?
> 2. Given that STELLA relies heavily on structured protein data, how does it perform when dealing with less common protein structures?
> 3. In my experience, we often work with proteins that lack precise structure or even sequence information in certain regions. How well could STELLA handle incomplete protein data?
> 4. Does STELLA show a significant enough improvement over other established models to justify its scientific meaning and contribution?
> 5. I’m curious about the model’s scalability because of the interactive demonstration. Can STELLA efficiently handle large-scale datasets or high-throughput predictions?
>
> **Responses to Q1**
> - We understand the critical importance of accurate protein function interpretation, especially when applied to biotechnological research. The potential consequences of false interpretations—such as leading research in the wrong direction or wasting resources—are indeed significant. While STELLA provides valuable insights into protein function predictions, we agree that incorporating confidence measures and uncertainty quantification in predictions is essential. This could include mechanisms for signalling the reliability of predictions, based on model confidence or uncertainty in the embeddings, as well as validating predictions through experimental feedback. We plan to explore these directions in future iterations of STELLA to improve the robustness and trustworthiness of the generated predictions and to make the STELLA's reasoning process even more transparent and accessible to domain experts.
>
> **Responses to Q2**
> - In the function prediction (FP) task, we employed a rigorous data splitting strategy based on sequence similarity, ensuring that the similarity between training and test sequences was below 40%. This approach was designed to simulate scenarios involving rare or uncommon proteins, providing a more realistic evaluation of STELLA's ability to generalize to novel and previously unseen cases.
>
> **Responses to Q3**
> - Thank you for highlighting this important consideration, as incomplete protein data is indeed a common challenge in protein biology research. To address this, we conducted an additional experiment to evaluate STELLA’s ability to handle incomplete protein structures. Specifically, for the testing data, we cut away the terminal 10% of the protein structures to simulate incomplete structural information and assess the model’s performance under these conditions. STELLA's performance, measured by the ROUGE-L score, shows a slight decrease from 0.5257 to 0.4915. Considering that the training was conducted using complete protein structures, this slight decline due to inconsistency still demonstrates the robustness of our model, indicating its capability to handle incomplete protein information in application scenarios.

---

> > ### Comment · Reviewer_qSap · 2024-11-26
> > **Response to Authors**
> >
> > Thank you for the response.
> > The explanation to Q2 is acceptable and addresses the concern sufficiently. The response to Q3 thoroughly resolves my concerns and significantly strengthens the argument presented in the paper.

---

> ### Author Response · Authors · 2024-11-25
> **Response to Reviewer qSap -- #2**
>
> **Questions:**
>
> 4. Does STELLA show a significant enough improvement over other established models to justify its scientific meaning and contribution?
> 5. I’m curious about the model’s scalability because of the interactive demonstration. Can STELLA efficiently handle large-scale datasets or high-throughput predictions?
>
> **Response to Q4**
> - Sequence, structure, and function (text) are three core data modalities in protein biology. Structural data, in particular, plays a crucial role in uncovering biological functions. Despite the extensive structural data which are publicly available, such as the PDB and AFDB, there is still a pressing challenge to fully utilize these resources to delve into understanding of protein functions. Bridging the gap between structural data and functional or biochemical insights is essential to unlock their full potential for practical applications in both research and industry.
> - To address this challenge, innovative approaches that integrate structural data with cutting-edge computational tools are urgently needed.  STELLA represents a transformative solution, combining the strengths of LLMs with protein structure insights to deliver accurate, versatile, and interactive predictions across a wide range of protein-related tasks (as demonstrated in Appendix A.1 with Examples 1 and 2 in the revised manuscript). By achieving state-of-the-art performance in multiple tasks, STELLA demonstrates the potential of multimodal LLMs as a groundbreaking paradigm for advancing protein biology research. This advancement not only builds upon but also transcends the capabilities of traditional protein language models (pLMs), as highlighted by reviewers tXk1, cosX, and L7ds, who recognized the effective utilization of LLM capabilities as a key strength of this work.
> - STELLA introduces a novel methodology for understanding proteins beyond the existing pLM frameworks and expands the boundaries of LLM applications in protein biology. This study paves the way for a new paradigm in protein science and computational biology, offering a more integrated and efficient approach to leveraging structural data for scientific discovery.
>
> **Response to Q5**
> - Scalability is a critical consideration for STELLA, especially given its interactive demonstration capabilities. Our model is designed with scalability in mind, enabling it to efficiently handle large-scale datasets and perform high-throughput predictions. Incorporating optimization techniques and leverage state-of-the-art frameworks such as vLLM[1] helps to accelerate inference. By leveraging the computational efficiency of LLMs and implementing optimization techniques such as batching and parallel processing, STELLA maintains robust performance even with increasing data volume. These optimizations can make STELLA a robust and efficient tool, suitable for both interactive exploration and high-throughput scientific workflows, paving the way for broader applications in protein science.
>   > [1] Kwon, W., Li, Z., Zhuang, S., Sheng, Y., Zheng, L., Yu, C.H., Gonzalez, J., Zhang, H. and Stoica, I., 2023, October. Efficient memory management for large language model serving with pagedattention. In Proceedings of the 29th Symposium on Operating Systems Principles (pp. 611-626).

---

> > ### Comment · Reviewer_qSap · 2024-11-26
> > **Response to Authors**
> >
> > Thank you for the response. As noted by other reviewers, the unique contributions and advances of this work should be emphasized more clearly. Doing so would enhance the overall impact and help better position the work within the field.

---

### Official Review · Reviewer_tXk1 · 2024-11-03

**Soundness:** 3
**Presentation:** 4
**Contribution:** 3
**Rating:** 8
**Confidence:** 4

**Summary:**

STELLA offers a valuable protein function prediction tool powered by large language models, potentially having a significant impact on the field of bioinformatics. Key contributions include the possibility of predicting protein function using structure information and advanced LLM capabilities, demonstrating STELLA’s effectiveness in function prediction. STELLA can facilitate research in life science and has the potential to provide another layer of information besides folding structures like AlphaFold.

**Strengths:**

Overall, this submission is structured clearly and defines the biological question it aims to address. STELLA  originality is based on its innovative approach to bridging structural representations with LLM capabilities, which allows it to interpret complex protein structures and respond to diverse contextual queries. This provides another layer of protein functionality besides its folding structure like emerging popular tools such as AlphaFold. This submission demonstrates solid technical foundation such as showing results based on a two-stage multimodal instruction tuning process, combinations trying of models, and comparable improvements from previous similar SOTA models.

**Weaknesses:**

The metrics used in the evaluation section may not fully capture the biological relevance and accuracy of protein function predictions (just comparing generated answer vs ground truth description of protein function). The BLEU score, for instance, may not reflect nuanced but critical differences in function. Moreover, I am quite concerned about the limitation of the OPI-Struc dataset. Although the authors mentioned these target on Function and Enzyme, it is better to evaluate the tasks based on more diverse types of classification of proteins (functional protein used as virulence factors in bacteria vs functional protein in mitochondria in mice). How the future impacts of STELLA, especially how confident the interpretation of the protein, should be further discussed. The false interpretation of protein function may lead to the wrong direction of biotech operation, which results in a significant waste of funding.

**Questions:**

1. Could the authors share insights on why ESM3 was preferred over other potential encoders, this will provide reasons for future studies to choose ESM3 as a protein encoder.
2. Could the authors provide clarification on why STELLA performs comparably to Prot2TextLARGE in some metrics but underperforms in others (e.g., ROUGE and BERT-score for specific tasks)? Adding an explanation would be helpful to understand the specific cases or factors influencing these results.
3.. On Table 3, since ESM3, and Prot2Text these models cannot be evaluated using acc@MCQA_1x and acc@MCQA_4x, is it worth putting these results into a table?
4. A few sentences describing future user usage will be appreciated. How do researchers in the science field use STELLA and more importantly how confident the results from STELLA can guide the direction of research and even decision formation?

---

> ### Author Response · Authors · 2024-11-25
> **Response to Reviewer tXk1**
>
> Thank you for raising 3 insightful questions for our work, which are addressed below.
>
> **Response to Q1**
> - Thank you for giving this construction suggestion. ESM3 underwent self-supervised learning in masked language modeling (MLM) mechanism using massive proteins across three modalities—sequence, structure, and function keywords—at the residue level. This residue-level multimodal interaction allows ESM3 to learn more fine-grained protein representations. In Figure 4, we provide a UMAP visualization showing ESM3's better representation ability than other protein encoders. Further details on the encoder can be found in Table 9. Also, ESM3 is widely recognized and utilized in the computational biology community, providing extensive validation and benchmarking. Its widespread use ensures that results can be easily compared and contextualized with other studies. In conclusion, ESM3’s balance of scalability, versatility, and high-quality representation learning makes it a compelling choice for protein-related research. We hope that these insights will assist future studies in selecting appropriate encoders for their specific objectives.
>
> **Response to Q2**
> - Thank you for your insightful question. We observed improvements across all metrics by extending the stage-2 training from 3 to 6 epochs. **As a result, STELLA outperforms Prot2Text$_{LARGE}$, achieving state-of-the-art performance in the FP task.**
>
> > **Evaluation results of the FP task, comparing with existing work.**  **Bold** and _italic_ indicate the best and the runner-up performance.
>
> >| Model/Method                            | BLEU-4 ↑ | BERT Score ↑ | ROUGE-1 ↑ | ROUGE-2 ↑ | ROUGE-L ↑ |
> |-----------------------------------------|-----------|--------------|------------|------------|------------|
> | Prot2Text$_{BASE}$ [abdine2023prot2text] | 0.3511    | 0.8430       | 0.5059     | 0.4271     | 0.4849     |
> | Prot2Text$_{LARGE}$ [abdine2023prot2text] | 0.3629    | _0.8520_     | _0.5368_   | _0.4560_   | _0.5140_   |
> | STELLA-ESM3-Llama-3.1-8B-Instruct (e3+e3) | _0.4024_  | 0.8496       | 0.5218     | 0.4487     | 0.5041     |
> | STELLA-ESM3-Llama-3.1-8B-Instruct (e3+e6) | **0.4300**| **0.8564**   | **0.5423** | **0.4747** | **0.5257** |
> | Foldseek + Manual retrieval            | 0.3627    | 0.8358       | 0.4799     | 0.4027     | 0.4586     |
>
>
> - Although we have achieved SOTA performance with the above metrics in the FP task, we acknowledge that given the specialization and complexity of biological function descriptions, the quality of LLM responses cannot be fully captured by a single NLP metric. Therefore, we employ multiple metrics above to provide a more comprehensive evaluation of STELLA’s effectiveness. Additionally, recognizing the limitations of such conventional NLP metrics in protein-related tasks, we intentionally designed multiple-choice QA (MCQA) tasks to objectively assess STELLA’s performance.
> - Regarding the concern about Table 3, to enhance clarity and conciseness, we have decided to remove the table and simplify the content for improved readability and flow.
>
> **Response to Q3**
> - Thank you for your constructive suggestions. We list several potential user usage scenarios and will include additional dialogue examples in the revised version of the manuscript to demonstrate these capabilities:
>   - Providing comprehensive clarifications and addressing specific questions related to terminologies or concepts in biological processes.
>   - Highlighting similarities and differences among proteins involved in the same biological process. (see example 1 in Appendix A.1 of the revised manuscript)
>   - Suggesting novel biotechnological applications based on protein functionalities. (see example 2 in Appendix A.1 of the revised manuscript)
>   - Offering iterative functional predictions according to domain experts' feedback or new experimental evidence.
> - To sum up, scientific research is a long and iterative process of trials and errors. STELLA showcases transformative potential by creating a new LLM-based paradigm besides the protein language model (PLM) paradigm for protein biology research. STELLA is expected to act as a research copilot by offering efficient feedback, supporting continuous improvements, and empowering domain experts to make more informed decisions through dynamic, iterative interactions.

---

> > ### Comment · Reviewer_tXk1 · 2024-11-26
> > **Sufficient and useful response**
> >
> > Thank you for the response, which addressed my concerns sufficiently.

---

### Official Review · Reviewer_L7ds · 2024-11-03

**Soundness:** 2
**Presentation:** 2
**Contribution:** 2
**Rating:** 3
**Confidence:** 3

**Summary:**

This paper presents a multimodal model, STELLA, which integrates the tertiary structure of proteins into LLM, enabling LLM to understand PDB data to some extent. Through natural language instructions, STELLA can provide information about the protein in the input PDB. The model was trained and tested on the OPI-Struc dataset, achieving comparable results in function prediction and enzyme prediction.

**Strengths:**

The proposed model can conduct multi-round dialogues, presenting potential advantages in the interaction between experts and machines.

**Weaknesses:**

1. The authors state: "However, the PDB entries still lack detailed functional annotations except for function keywords." To my knowledge, each PDB co-crystal structure comes from a high-level research paper, which contains detailed research on the protein in the PDB. I am unsure what the authors mean by "lack detailed functional annotations."

2. The authors state: "these methods rarely incorporate iterative feedback from domain experts, a critical factor for refining predictions and improving their accuracy." While STELLA does support multi-round dialogues, it does not achieve optimal performance in Function Prediction (FP) and Enzyme Name Prediction (EP) tasks, showing no advantage over other methods that do not support multi-round dialogues. This seems to contradict the authors' claim.

3. The authors state: "Traditional methods often struggle to integrate the fine-grained structural details needed for accurate enzyme prediction, particularly when trying to model the influence of both local and global structural factors." Firstly, STELLA does not outperform the so-called "Traditional methods" in the EP task. Secondly, STELLA does not seem to mention how it handles "local and global structural factors."

4. Prot2Text is a baseline in this paper. Compared to Prot2Text, STELLA lacks sequence information, and the protein structure encoder and LLM backbone are different, but there is not much difference otherwise. I would like to know why the sequence information was removed. Moreover, as shown in Table 2, STELLA does not have an advantage over Prot2Text in the FP task.

5. As shown in Table 5, STELLA does not perform well in the EP task, falling below CDConv and New IEConv.

6. Figure 3 and Figure 4 do not seem to provide any useful information. Additionally, the authors use "Fig. X" when referencing figures, but the caption of the figures is "Figure. X.

**Questions:**

See Weaknesses.

---

> ### Author Response · Authors · 2024-11-25
> **Response to  Reviewer L7ds**
>
> Thank you for raising several insightful questions (weaknesses) for our work, which are addressed below.
>
> **Response to Q1**
> - Thank you for highlighting this point. While it is true that PDB structures are derived from detailed research publications, the annotations in the PDB typically include only abstracts and keywords, which do not provide the comprehensive, expert-reviewed functional descriptions found in databases like Swiss-Prot. We have revised the phrasing in the manuscript to avoid any potential misunderstandings. Meanwhile, we want to emphasize the lack of functional annotations for proteins in the AlphaFold Protein Structure Database and to further enable exploration of the protein universe in which many of our proteins remain hidden.  This work aims to develop new AI methods to discover the unknown knowledge of proteins. We appreciate your feedback, as it helps us refine and clarify our motivation more effectively.
>
> **Response to Q2**
> - **We achieved state-of-the-art (SOTA) results** in both the FP and EP tasks, with a **ROUGE-L of 0.5257 in the FP task** and an **accuracy of 0.8885 in the EP task**. Notably, in the FP task, the ROUGE-L score improved from 0.5041 to 0.5257 as we increased the stage-2 training epochs from 3 to 6. Similarly, in the EP task, the accuracy increased from 0.8806 to 0.8885 with the same adjustment in training epochs.
>
> **Response to Q3**
> - Thank you for your insightful feedback. We acknowledge that our original phrasing in the introduction may have unintentionally misled readers regarding the handling of “local and global structural factors.” To address this, we have revised the statement in the introduction to clarify our intent. Specifically, we aim to emphasize that enzyme function prediction often correlates with certain structural regions, such as active sites or binding pockets, rather than implying that our framework explicitly models local and global structural factors.
> - Our focus in this work is on leveraging multimodal modeling to integrate structural information in a more general sense, without explicitly isolating or addressing these factors. We believe this clarification better aligns with the design and scope of STELLA and avoids potential misinterpretation. Thank you for helping us improve the clarity of our manuscript.
>
> **Response to Q4**
> - Thank you for your thoughtful question. First, we would like to clarify that STELLA does not completely remove sequence information. Specifically, the ESM3 encoder takes residue-level sequence information as its partial input.  However, our study emphasizes the utilization of structural data as a complementary modality to sequence information, aiming to explore the potential of multimodal modeling based on structures in protein biology. This deliberate focus allows us to investigate how structural insights can enhance predictions and address challenges that sequence-based models alone may not fully resolve. We achieved SOTA results in the FP task, with a ROUGE-L = 0.5257，surpassing previous ROUGE-L=0.5140 of Prot2Text_Large.
>
> **Response to Q5**
> - We achieved a SOTA result with an accuracy of 0.8885 in the EP task by increasing the stage-2 training epochs to 6. Given the limited size of the Enzyme dataset, extending the training from 3 to 6 epochs allowed the model to better learn from the available data. This led to an improvement in accuracy from 0.8806 to 0.8885, achieving SOTA performance.
>
> **Response to Q6**
> - Thank you for your valuable suggestion. We have improved the clarity of the paper’s presentation. In response, we would like to clarify the roles of Figures 3 and 4: Figure 3 illustrates the length distribution of protein sequences in the dataset, highlighting the model’s applicability to functional descriptions across proteins of varying lengths. Besides the length distribution, we have also provided analysis of data label distribution of our OPI-Struc dataset in the Appendix. These analyses aim to offer a comprehensive understanding of the dataset’s characteristics. These efforts confirm that our dataset has been meticulously curated to  provide a high-quality resource for the community, support robust model evaluation, and offer guidance for constructing future datasets. **This attention to details reflects our commitment to advancing LLM-driven paradigm for protein biology research and laying a dependable foundation for future scientific endeavours.**
> - Figure 4 emphasizes the effectiveness of the three encoders in modeling proteins from the perspective of function annotation. Among them, ESM3 demonstrates superior performance, consistent with the model’s results on both the FP and EP tasks. Additionally, we have updated the referencing format for the figures as suggested.

---

> > ### Comment · Reviewer_L7ds · 2024-11-26
> >
> > Thank you for the response. I have three main concerns:
> >
> > + **What are the practical applications of STELLA?** Could the author provide specific examples of information that cannot be obtained from the PDB, but can be obtained accurately, or with very high confidence, through STELLA?
> >
> > + **STELLA did not perform as well as expected**, and did not demonstrate the superiority claimed by the author. Based on your response, have you conducted additional experiments that achieved better results than the initial submission?
> >
> > + **The novelty and originality of this work.** Please discuss Prot2Text[1] and ProteinGPT[2], detailing the differences between STELLA and them, as well as its superiority.
> >
> > [1] Abdine, Hadi, et al. "Prot2text: Multimodal protein’s function generation with gnns and transformers." Proceedings of the AAAI Conference on Artificial Intelligence. Vol. 38. No. 10. 2024.
> >
> > [2] Xiao, Yijia, et al. "Proteingpt: Multimodal llm for protein property prediction and structure understanding." arXiv preprint arXiv:2408.11363 (2024).

---

> > > ### Author Response · Authors · 2024-11-26
> > > **Response to Reviewer L7ds (Follow-up concerns) -- #1**
> > >
> > > **Response to follow-up Q1**
> > > - Thank you for raising this important concern. We acknowledge that the practical applications of LLMs and multimodal LLMs are a critical area of focus, **not only in the domain of protein biology but also in fields such as NLP and computer vision**. This aspect is one we deeply value and aim to contribute to meaningfully.  We firmly believe that advancing the practical applications of such methodologies **requires the collective input of peer reviewers and domain experts to refine and mature innovative approaches**. Sharing STELLA with the broader community through publication aligns with our vision of not merely achieving SOTA performance but also **push the frontier of LLM-based methodology in protein biology, a promising research direction**. The evolution of LLM-based methodologies for practical applications lies in progressively enhancing their capabilities, **from knowledge retrieval to reasoning, and finally to task orchestration**. These advancements enable LLMs to tackle more complex, real-world application scenarios. **Achieving this progress demands collaborative efforts from academia and industry to fully realize the potential of this approach**.
> > > -  As the Reviewer CVwt noted,  "While such multimodal integration is common in vision and NLP, there’s been virtually no work in the space of protein structure and NLP (function), which this paper pioneers. The core idea of using a structure-based embedding to translate protein structures into a common latent space with NLP-based function annotations is clever and is sufficiently interesting and promising that it may become a whole new research direction." **This recognition highlights STELLA’s innovation and its significant contribution to the community.**
> > > - In addition to achieving SOTA performance in downstream tasks, we have explored STELLA’s potential in practical applications, demonstrating its versatility in addressing complex scenarios such as:
> > >   - Protein comparison: Highlighting similarities and differences among proteins involved in the same biological process. (see **example 1**, which is included in Appendix A.1 of the manuscript)
> > >   - Application proposals: Suggesting novel biotechnological applications based on protein functionalities. (see **example 2**, which is included in Appendix A.1 of the manuscript)
> > >   - Explanation of biological processes: Providing comprehensive clarifications and addressing specific questions related to terminologies or concepts in biological processes. (see the **example** in Section 3 of the manuscript)
> > >
> > > **Example 1**
> > > ```
> > > > User: What are the main functions of this protein?[AFDB_ID: AF-Q9W3K5-F1-model_v4.pdb]
> > > > STELLA: The main functions of this protein are to catalyze the ATP-dependent ligation of L-glutamate and L-cysteine, and to a lesser extent, of L-histidine and L-tryptophan.
> > > > User: Compare the function of glutamate-cysteine ligase with that of glutathione synthetase. What structural or mechanistic features distinguish these enzymes, and how do they together regulate glutathione levels?
> > > > STELLA: Glutamate-cysteine ligase and glutathione synthetase are both involved in the synthesis of glutathione, a critical antioxidant in cells. However, they have different functions and mechanisms of action. Glutamate-cysteine ligase catalyzes the ligation of L-glutamate and L-cysteine to form a peptide bond, while glutathione synthetase catalyzes the condensation of L-glutamate and L-cysteine to form glutathione. The two enzymes have different substrate specificities and reaction conditions, and they work together to regulate glutathione levels in cells.
> > > ```
> > > **Example 2**
> > > ```
> > > > User: What are the main functions of this protein?
> > > [AFDB_ID: AF-Q5KYR2-F1-model_v4.pdb]
> > > > STELLA: The main functions of this protein are involved in the isomerization of 5-deoxy-glucuronate (5DG) to 5-dehydro-2-deoxy-D-gluconate (DKG or 2-deoxy-5-keto-D-gluconate).
> > > > User: Describe potential industrial or medical applications of the isomerization catalyzed by this protein, specifically converting 5-deoxy-glucuronate (5DG) to 5-dehydro-2-deoxy-D-gluconate (DKG). How might this reaction be utilized in biotechnology or pharmaceutical production?
> > > > STELLA: The isomerization reaction catalyzed by this protein has several potential applications in biotechnology and pharmaceutical production. For example, the conversion of 5DG to DKG could be used to produce novel compounds or to improve the efficiency of existing ones. Additionally, the reaction could be used to develop new biotechnological processes or to improve existing ones.
> > > > User: Any further examples?
> > > > STELLA: Yes, the isomerization reaction could also be used to produce novel antibiotics, anticancer agents, or other bioactive compounds. For example, the conversion of 5DG to DKG could be used to synthesize novel peptidoglycan structures, which are important for bacterial cell wall synthesis and could potentially be used to develop new antibiotics.
> > > ```

---

> > > ### Author Response · Authors · 2024-11-26
> > > **Response to Reviewer L7ds (Follow-up concerns) -- #2**
> > >
> > > **Response to follow-up Q2**
> > > - We are pleased to report that we have **achieved SOTA performance in both the function prediction (FP) and enzyme prediction (EP) tasks, with ROUGE-L = 0.5257 and accuracy = 0.8885**, respectively.
> > > - We understand that you may be curious about the details of the improvements. In our initial submission, we conducted extensive comparative experiments. However, due to the shortage of GPU resource, we capped the stage-2 training at a maximum of 3 epochs for all experiments. To address this, currently we conducted an additional experiment where stage-2 training was extended to 6 epochs, starting from scratch. **In the additional experiments, all hyperparameters and experimental setups remained consistent with the initial submission, while the only adjustment is the increased training epochs in stage 2.**
> > > - As a result, our additional experiments demonstrated the SOTA performance of STELLA in both FP and EP tasks, surpassing previous respective benchmarks, including Prot2Text_Large (ROUGE-L = 0.5140) for the FP task and CDConv (Accuracy = 0.8850) for the EP task.
> > > - **The final experimental results significantly highlight that STELLA outperforms existing approaches, which we think have addressed your concerns in the previous review, i.e., Q4 & Q5.**
> > >
> > > > **Table: Evaluation results of the FP task, comparing with existing work.**  **Bold** and _italic_ indicate the best and the runner-up performance.
> > >
> > > >| Model/Method                            | BLEU-4 ↑ | BERT Score ↑ | ROUGE-1 ↑ | ROUGE-2 ↑ | ROUGE-L ↑ |
> > > |-----------------------------------------|-----------|--------------|------------|------------|------------|
> > > | Prot2Text$_{BASE}$ [abdine2023prot2text] | 0.3511    | 0.8430       | 0.5059     | 0.4271     | 0.4849     |
> > > | Prot2Text$_{LARGE}$ [abdine2023prot2text] | 0.3629    | _0.8520_     | _0.5368_   | _0.4560_   | _0.5140_   |
> > > | STELLA-ESM3-Llama-3.1-8B-Instruct (e3+e3) | _0.4024_  | 0.8496       | 0.5218     | 0.4487     | 0.5041     |
> > > | STELLA-ESM3-Llama-3.1-8B-Instruct (e3+e6) | **0.4300**| **0.8564**   | **0.5423** | **0.4747** | **0.5257** |
> > > | Foldseek + Manual retrieval            | 0.3627    | 0.8358       | 0.4799     | 0.4027     | 0.4586     |
> > >
> > > > **Table: Evaluation results of the EP task, comparing with existing work.**  **Bold** and *italic* indicate the best and runner-up performance.
> > >
> > > > | Model                                           | Training Manner | Acc@EP ↑     |
> > > |------------------------------------------------|-----------------|--------------|
> > > | DeepFRI (Gligorijevi´c et al., 2021)                              | w/ pretrain     | 0.6330       |
> > > | UniRep (Alley et al., 2019)                              | w/o pretrain    | 0.7290       |
> > > | 3DCNN (Derevyanko et al., 2018)                              | w/o pretrain    | 0.7880       |
> > > | HH-suite3 (Steinegger et al., 2019)                           | w/o pretrain    | 0.8260       |
> > > | ESM-1b (Rives et al., 2021)                            | w/ pretrain     | 0.8310       |
> > > | GearNet-Edge-IEConv  (Zhang et al., 2022)                  | w/o pretrain    | 0.8530       |
> > > | IEConv(Hermosilla et al., 2021)                           | w/o pretrain    | 0.8720       |
> > > | GearNet-Multiview-Contrast (Zhang et al., 2022)            | w/ pretrain     | 0.8750       |
> > > | New IEConv (Hermosilla and Ropinski, 2022)                        | w/ pretrain     | 0.8810       |
> > > | CDConv  (Fan et al., 2022)                        | w/o pretrain    | *0.8850*     |
> > > | **STELLA-ESM3-Llama-3.1-8B-Instruct**          | MMIT            |              |
> > > | - (single, two-stage, e3+e3)                   |                 | 0.8806       |
> > > | - (single, two-stage, e3+e6)                   |                 | **0.8885**   |

---

> ### Author Response · Authors · 2024-11-26
> **Response to Reviewer L7ds (Follow-up concerns) -- #3**
>
> **Response to follow-up Q3**
>
> - Thank you for raising this concern and giving suggestions to discuss Prot2Text[1] and ProteinGPT[2]. This allows us to clarify the unique advantages of STELLA.
>
> - **Prot2Text**
>
>   - Prot2Text employs a graph-based encoder and a GPT-2 decoder, designed to take protein structures as input and generate functional annotations as output. To ensure a fair comparison, we conducted experiments using the exact same dataset splits as Prot2Text for the FP task. Notably, this dataset split is highly rigorous, **maintaining sequence identity < 40% between the training and testing sets, which we recognize as a significant contribution of Prot2Text**.  Additionally, **we integrated Prot2Text’s encoder into our model architecture as part of an ablation study exploring different protein encoders and LLMs, as detailed in Section 5.3.2**. Under these conditions and through extensive experiments, STELLA achieved SOTA performance, highlighting **its clear advantages over Prot2Text** in terms of accuracy and versatility.
>   - While Prot2Text is a specialized model focused on static protein function prediction, it has inherent limitations in meeting the broader and dynamic needs of domain experts. In contrast, STELLA operates as a multimodal LLM capable of generating interactive functional annotations in a conversational format. This capability enables more dynamic and user-centric interactions. Moreover, **STELLA goes beyond functional annotation by supporting additional tasks**, such as enzyme prediction, which underscores its **versatility and adaptability** to diverse research needs.
>
> - **ProteinGPT**
>
>   - While ProteinGPT is also a multimodal LLM, STELLA distinguishes itself through differences in dataset construction and model evaluation:
>   - **Dataset construction**:
>     - **ProteinGPT leverages abstract descriptions from the RCSB-PDB** database as the text modality, while **our OPI-Struc dataset is built using detailed annotations from the Swiss-Prot database**. Swiss-Prot provides more comprehensive protein functional annotations curated and reviewed by domain experts, ensuring higher data quality and greater reliability for downstream tasks.
>
>   - **Model evaluation**:
>     - **STELLA takes a more strict evaluation strategy with rigorous dataset split**.  **ProteinGPT employs 7:3 split ratio** for dividing the training and testing sets. In contrast, **STELLA adopted a more stringent** data splitting strategy, ensuring **sequence similarity between training and test sets is below 40%**. **The dataset split strategy for STELLA better reflects real-world scenarios for annotating uncommon proteins.**
>     - **STELLA takes protein-related specialized models  to compare for more strict and responsible evaluation.** Regarding **the choice of baseline models for comparison**, **ProteinGPT selected general-purpose LLMs that are not specifically optimized for protein-related tasks**. In our work, we compared **STELLA against previous SOTA specialized models** for both the function prediction (FP) and enzyme prediction (EP) tasks (e.g., Prot2Text_LARGE and CDConv), establishing a more strict and responsible evaluation.
> - Overall, the evaluation strategy in our work ensures rigorous, reasonable and reliable assessment of STELLA’s performance, which is also recognized by Reviewer CVwt as one of our work’s strengths.

---

> ### Comment · Reviewer_L7ds · 2024-11-26
>
> **Regarding Q1**, you have consistently failed to directly answer my question. Since you claim that your model can obtain information that is not available in the PDB database, I only ask you to provide one or two examples. Moreover, proposing a model and then asking the community to verify its broader applications does not seem to be a responsible approach. Furthermore, the novelty of your model is not apparent. Don't mention Reviewer CVwt to me, I question his professionalism. This work is clearly very similar to Prot2Text and ProteinGPT that I mentioned, and he irresponsibly gave a score of 10 without thorough investigation, which makes me question his motives. Also, Reviewer CVwt himself mentioned about your initial model version that "The model does not really push the state of the art" and is "missing technical details". I maintain a high level of skepticism about his high score of 10 under these circumstances.
>
> **Regarding Q2**, I can accept that you achieved better performance after further training, although the improvement is quite limited, such as an increase of only 0.0035 on EP. Such a negligible improvement could be invalidated due to various factors, such as data partitioning and sampling strategies. However, this still clearly does not sufficiently demonstrate the role of 'iterative feedback.'
>
> **Regarding Q3**, what is the fundamental difference between your model and Prot2Text and ProteinGPT? The difference between STELLA and Prot2Text is merely a change in the encoder and LLM backbone; you also said: "STELLA distinguishes itself through differences in dataset construction and model evaluation". I am curious why you brought up Reviewer CVwt again; I am almost certain that his/her review comments hold little reference value.

---

> > ### Author Response · Authors · 2024-11-27
> > **General Response to Reviewer L7ds (Follow-up concerns - R2 )**
> >
> > - Thank you again for your detailed feedback and for raising these important concerns. We apologize if our previous response, in referencing Reviewer CVwt’s comments, gave the impression of diminishing the significance of your review or causing any misunderstanding. That was not our intention at all. Our sole purpose in citing other reviewers’ feedback was to provide a more comprehensive and balanced response.
> > - We deeply value your thoughtful input and recognize its critical role in improving the clarity and rigor of our work. Your feedback has significantly contributed to shaping our work and its future direction.
> > - Once again, we appreciate your time and effort in evaluating our work and remain committed to improving it further based on your constructive feedback. Moving forward, we will focus on addressing your specific concerns with technical arguments and evidence.

---

> ### Author Response · Authors · 2024-11-30
> **Additional Response to Reviewer L7ds (Follow-up concerns - R2 )**
>
> Dear Reviewer L7ds,
>
> Thank you once again for your detailed and thoughtful feedback. Your comments have been instrumental in leading us to refine our work further.  We also appreciate the opportunity to address your prior follow-up concerns.
> ******
>
> **Regarding Q1**
> - We sincerely apologize for any misunderstanding caused by the presentation that 'STELLA can obtain information that is not available in the PDB database' in our initial submission. **To clarify, STELLA does not possess the capability to obtain information that is entirely unavailable in the PDB database**. Instead, its strengths lie in leveraging structure-based representations to address two protein-related tasks as follows:
>   - Functional description prediction. Using the structural data available in the AFDB, STELLA can help predict functions of proteins whose structures are known but whose biological functions remain unclear. This functionality is expected to be valuable for exploring unannotated proteins.
>   - Enzyme-catalyzed reactions prediction. STELLA can utilize structural data to predict enzyme-catalyzed reactions, providing insights into potential catalytic activities based on structural features.
>
> **Regarding Q2**
> - Thank you for pointing out the need to clarify the role of 'iterative feedback' in our evaluation. Iterative feedback refers to the process that LLMs improve and adjust its responses based on previous outputs or additional user inputs. **However, we acknowledge that quantifying this capability is currently challenging, as existing benchmarks for multimodal LLMs are not designed to assess dynamic and iterative feedback. Our quantitative evaluation of STELLA focused on static performance metrics, such as ROUGE-L and accuracy, which do not capture the iterative nature of this process.**  In our current quantitative evaluation of STELLA, we conducted only a single round of interaction for each task.  We regret not adequately explaining these limitations in our original submission. Thank you again for your insightful review to help us identify areas for improvement.
>
> **Regarding Q3**
>
> - STELLA adopts a projection-based multimodal LLM architecture, inspired by widely recognized vision-language models like LLaVA[1]. The prior works such as Prot2Text[2] and ProteinGPT[3] also unitize a projection layer (e.g., MLP or Linear layers) to build the multimodal framework. These models combine embeddings from the biomolecular modality (e.g., sequences or structures, or their fusion) with the text modality, aiming to leverage the capabilities of LLMs to handle biomolecular tasks.
>
> - The fundamental difference between STELLA, Prot2Text and ProteinGPT is the modality fusion strategy. **Prot2Text and ProteinGPT integrate protein sequence and structure data using a late fusion strategy**, where each modality is encoded separately before being fused. However, late fusion approaches have certain limitations, such as the potential loss of cross-modal relationships and increased complexity of encoder modules. **In contrast, STELLA adopts an early fusion strategy inherited from the ESM3 encoder**, where the sequence and structure modalities are jointly represented and fused into a unified representation at encoding stage. This early fusion strategy has the potential to both preserve the intrinsic relationships between modalities and improve computational efficiency.
>
> > [1] Liu, H., Li, C., Wu, Q. and Lee, Y.J., 2024. Visual instruction tuning. Advances in neural information processing systems, 36.
> > [2] Abdine, Hadi, et al. "Prot2Text: Multimodal protein’s function generation with gnns and transformers." Proceedings of the AAAI Conference on Artificial Intelligence. Vol. 38. No. 10. 2024.
> > [3] Xiao, Yijia, et al. "ProteinGPT: Multimodal llm for protein property prediction and structure understanding." arXiv preprint arXiv:2408.11363 (2024).
>
> Thank you for highlighting the important issues. We have revised the manuscript to present STELLA’s capabilities and limitations more precisely. Your feedback is greatly valued. We hope this revision effectively highlights STELLA’s unique features. Please let us know if further clarification is needed. Thanks a lot.
> ******
> Best regards,
> STELLA Team

---

### Official Review · Reviewer_CVwt · 2024-11-03

**Soundness:** 4
**Presentation:** 3
**Contribution:** 4
**Rating:** 10
**Confidence:** 3

**Summary:**

This paper introduces a new approach for protein function prediction by integrating proteins LLMs (e.g., ESM3) with NLP LLMs (Llama-3.1) to essentially “translate” from a structure-based representation into a natural language one. Part of this work involved creating a new dataset for training and evaluation.

**Strengths:**

* Integrating and translating between LLMs in entirely different domains, e.g., protein structure and natural language, is a new and highly promising direction. While such multimodal integration is common in vision and NLP, there’s been virtually no work in the space of protein structure and NLP (function), which this paper pioneers.
* To accomplish the authors create a new dataset, which they call Open Protein Instructions for Structures (OPI-Struc), a new effort into its own right to be able to train this model and assess it rigorously.
* The core idea of using a structure-based embedding to translate protein structures into a common latent space with NLP-based function annotations is clever and is sufficiently interesting and promising that it may become a whole new research direction.
* Evaluations are done rigorously, there’s not a tendency to try to inflate the results (great!), and some ablations are performed to assess different contributions to model performance.

**Weaknesses:**

* The paper largely builds on an existing framework for vision-NLP integration (LLaVA) by modifying it to the protein domain. Given how different proteins are from vision, it is likely that much further advancement can be had by innovating architecturally. However, this is a minor quibble as this paper pushes the frontier of protein multimodal integration and it makes sense to start with known architectures.
* The actual results are a bit underwhelming. The model does not really push the state of the art. Nonetheless, I consider this a minor issue as it introduces a new way of performing protein function prediction which I am sure can be improved substantially in the future.

**Questions:**

Please fill in some of the currently missing technical details, even if the code will be available. For instance what ESM3 model is used is not described (there are multiple).

---

> ### Author Response · Authors · 2024-11-24
> **Response to Reviewer CVwt**
>
> **Response to Q1**
> - Thank you for your insightful comments and constructive suggestions. We greatly appreciate your feedback, which not only highlights key areas for refinement and future exploration but also acknowledges the contributions of this work to the community.
> - As for the technical details, we utilized the 1.4B parameter version of ESM3 (esm3_sm_open_v1) in our experiments.  AFDB and PDB data were used as raw input, and embeddings were extracted using the official inference pipeline provided by the ESM3 framework. This ensured consistency, reproducibility, and alignment with established best practices.
> - We recognize the importance of providing clear and comprehensive technical details and will incorporate this clarification into the manuscript to enhance its transparency and accessibility. Once again, thank you for your valuable feedback and thoughtful suggestions!

---

### Official Review · Reviewer_cosX · 2024-11-04

**Soundness:** 3
**Presentation:** 3
**Contribution:** 3
**Rating:** 5
**Confidence:** 3

**Summary:**

This manuscript proposed STELLA, a multimodal LLM that integrates structural protein representations with LLMs to enhance protein understanding. It addresses the main limitation of traditional methods of solely depending on structural data and lacking the ability to incorporate iterative feedback from domain experts. This authors also developed the OPI-Struc dataset, conducted comprehensive evaluation, and provided open access to the code, datasets and models.

**Strengths:**

1. The proposed STELLA is an innovative approach that harnesses the capabilities of LLMs enriched with structural information. It has great potential to learn complex structure-function relationships from large datasets by integrating structural data with vast biochemical knowledge. It is also beneficial to bridge machine-readable protein language and human-readable natural language.

2. This paper present well-developed OPI-Struc dataset, and comprehensive evaluations. It takes into account the newer release of Swiss-Prot to assess the inference performance on unseen data, dataset options with or without permutations, and proper data split.

3. This paper is well written and well organized, with clear figure demonstration, tables, and good readability. It is easy for readers to follow.

**Weaknesses:**

1. For the important metrics, FP_{eval_FTQA(_v2401)} and EP_{eval}, STELLA performs slightly worse than the start-of-the-art methods. For metrics FP_{eval_MCQA}, although STELLA enables responding to this kind of questions, we lack baselines to demonstrate STELLA's superior performance. In addition, multiple-choice Q&A may not be a common use case for this model in practice.

2. There's a significant gap between metric FP_{eval_MCQA_1X} and metric FP_{eval_MCQA_4X}. It would be beneficial to include discussions and insights for this observation, and how to further reduce the sensitivity to the permutation.

3. (minor) This paper mentions the ability to incorporate iterative feedback from domain experts. Although this is only possible with the integration of LLM-based multi-turn dialogue (which we show in the paper), it might be useful to demonstrate this using some "expert-feedback" examples.

**Questions:**

Please kindly refer to section "Weaknesses".

---

> ### Author Response · Authors · 2024-11-24
> **Response to Reviewer cosX**
>
> Thank you for raising 3 insightful questions (weaknesses) for our work, which are addressed below.
>
> **Response to Q1&Q2:**
> - You are correct that MCQA is not commonly used in practical protein-related tasks. However, the purpose of designing this subtask is to serve as a supplementary evaluation method, providing a more comprehensive assessment of the model’s capabilities.  The specialized and complex nature of biological function descriptions makes **it challenging to fully capture the quality of LLM responses using conventional NLP metrics, such as BLEU, BERT and ROUGE scores**. Recognizing these limitations, **we intentionally included the MCQA subtask to objectively evaluate STELLA’s performance in a structured and quantifiable manner.**
> - For multiple-choice questions with four options, even random selection yields an accuracy rate of 25%, which reduces the performance differentiation between models. Furthermore, previous studies in multimodal LLMs revealed that LLMs may prefer to predict a certain choice among all given choices [1]. **Based on above considerations, we introduce FP_{eval_MCQA_4X} as a complementary evaluation metric besides FP_{eval_MCQA_1X}.**
> - When calculating FP_{eval_MCQA_4X}, the model is deemed successful in solving a question only if it correctly predicts the answer across all permutations. This requirement is significantly more challenging than answering a single question correctly. For instance, if a model randomly selects one of the four options, the probability of correctly answering a single question is 25%, whereas the probability of answering all permutations correctly drops drastically to  25\%^4 . **This substantial gap has also been reported in [1]： switching the evaluation from 1X to 4X results in a performance drop of -4.7% to -34.1% across 21 models, underscoring the increased difficulty and the stricter evaluation standard.**
>   > [1] MMBench: Is Your Multi-modal Model an All-around Player?
>
> **Response to Q3:**
> - Thank you for your interest in STELLA's ability to incorporate iterative feedback from domain experts. To further demonstrate this capability, we have provided two more examples below. In the example 1, STELLA responds to a domain expert's inquiry by comparing the similarities and differences in the functions of two proteins within the same biological process. This enables a deeper understanding of protein functionality. In the example 2, STELLA offers suggestions for biotechnological applications based on expert feedback, providing valid directions for further exploration.
>
> **Example 1**
> ```
> > User: What are the main functions of this protein?[AFDB_ID: AF-Q9W3K5-F1-model_v4.pdb]
> > STELLA: The main functions of this protein are to catalyze the ATP-dependent ligation of L-glutamate and L-cysteine, and to a lesser extent, of L-histidine and L-tryptophan.
> > User: Compare the function of glutamate-cysteine ligase with that of glutathione synthetase. What structural or mechanistic features distinguish these enzymes, and how do they together regulate glutathione levels?
> > STELLA: Glutamate-cysteine ligase and glutathione synthetase are both involved in the synthesis of glutathione, a critical antioxidant in cells. However, they have different functions and mechanisms of action. Glutamate-cysteine ligase catalyzes the ligation of L-glutamate and L-cysteine to form a peptide bond, while glutathione synthetase catalyzes the condensation of L-glutamate and L-cysteine to form glutathione. The two enzymes have different substrate specificities and reaction conditions, and they work together to regulate glutathione levels in cells.
> ```
> **Example 2**
> ```
> > User: What are the main functions of this protein?
> [AFDB_ID: AF-Q5KYR2-F1-model_v4.pdb]
> > STELLA: The main functions of this protein are involved in the isomerization of 5-deoxy-glucuronate (5DG) to 5-dehydro-2-deoxy-D-gluconate (DKG or 2-deoxy-5-keto-D-gluconate).
> > User: Describe potential industrial or medical applications of the isomerization catalyzed by this protein, specifically converting 5-deoxy-glucuronate (5DG) to 5-dehydro-2-deoxy-D-gluconate (DKG). How might this reaction be utilized in biotechnology or pharmaceutical production?
> > STELLA: The isomerization reaction catalyzed by this protein has several potential applications in biotechnology and pharmaceutical production. For example, the conversion of 5DG to DKG could be used to produce novel compounds or to improve the efficiency of existing ones. Additionally, the reaction could be used to develop new biotechnological processes or to improve existing ones.
> > User: Any further examples?
> > STELLA: Yes, the isomerization reaction could also be used to produce novel antibiotics, anticancer agents, or other bioactive compounds. For example, the conversion of 5DG to DKG could be used to synthesize novel peptidoglycan structures, which are important for bacterial cell wall synthesis and could potentially be used to develop new antibiotics.
> ```

---

> > ### Comment · Reviewer_cosX · 2024-11-27
> >
> > Thanks for the reply!
> >
> > I agree that adding the MCQA can provide a more comprehensive assessments, however, this contribution is more about enriching the evaluation instead of demonstrating that STELLA performs better. The significance of STELLA's contribution is still limited, given its limited performance on major metrics, and the innovation of model architecture and other aspects is also not significant enough to make up for it.

---

> > > ### Author Response · Authors · 2024-11-27
> > > **Response to Reviewer cosX (Follow-up concerns) -- #1 -- Regarding the performance on major metrics**
> > >
> > > **Response to follow-up concerns (regarding the performance on major metrics)**
> > > - Thank you for your agreement with the value of the MCQA as a complementary assessment of STELLA.
> > > - Also, thanks for raising the follow-up concerns which allow us to improve our work. **Regarding the performance on major metrics, we have achieved SOTA performance in both the function prediction (FP) and enzyme prediction (EP) tasks, with ROUGE-L = 0.5257 and accuracy = 0.8885**.  In our initial submission, we conducted extensive comparative experiments. However, due to the shortage of GPU resource, we capped the stage-2 training at a maximum of 3 epochs for all experiments. To address this, currently we conducted an additional experiment where stage-2 training was extended to 6 epochs, starting from scratch. In the additional experiments, all hyperparameters and experimental setups remained consistent with the initial submission, while the only adjustment is the increased training epochs in stage 2. As a result, in our additional experiments we achieved the SOTA performance of STELLA in both FP and EP tasks, surpassing previous respective benchmarks, **surpassing previous respective benchmarks, including Prot2Text_Large (ROUGE-L = 0.5140) for the FP task and CDConv (Accuracy = 0.8850) for the EP task**.
> > >
> > > > **Table: Evaluation results of the FP task, comparing with existing work.**  **Bold** and _italic_ indicate the best and the runner-up performance.
> > >
> > > >| Model/Method                            | BLEU-4 ↑ | BERT Score ↑ | ROUGE-1 ↑ | ROUGE-2 ↑ | ROUGE-L ↑ |
> > > |-----------------------------------------|-----------|--------------|------------|------------|------------|
> > > | Prot2Text$_{BASE}$ [abdine2023prot2text] | 0.3511    | 0.8430       | 0.5059     | 0.4271     | 0.4849     |
> > > | Prot2Text$_{LARGE}$ [abdine2023prot2text] | 0.3629    | _0.8520_     | _0.5368_   | _0.4560_   | _0.5140_   |
> > > | STELLA-ESM3-Llama-3.1-8B-Instruct (e3+e3) | _0.4024_  | 0.8496       | 0.5218     | 0.4487     | 0.5041     |
> > > | STELLA-ESM3-Llama-3.1-8B-Instruct (e3+e6) | **0.4300**| **0.8564**   | **0.5423** | **0.4747** | **0.5257** |
> > > | Foldseek + Manual retrieval            | 0.3627    | 0.8358       | 0.4799     | 0.4027     | 0.4586     |
> > >
> > > > **Table: Evaluation results of the EP task, comparing with existing work.**  **Bold** and *italic* indicate the best and runner-up performance.
> > >
> > > > | Model                                           | Training Manner | Acc@EP ↑     |
> > > |------------------------------------------------|-----------------|--------------|
> > > | DeepFRI (Gligorijevi´c et al., 2021)                              | w/ pretrain     | 0.6330       |
> > > | UniRep (Alley et al., 2019)                              | w/o pretrain    | 0.7290       |
> > > | 3DCNN (Derevyanko et al., 2018)                              | w/o pretrain    | 0.7880       |
> > > | HH-suite3 (Steinegger et al., 2019)                           | w/o pretrain    | 0.8260       |
> > > | ESM-1b (Rives et al., 2021)                            | w/ pretrain     | 0.8310       |
> > > | GearNet-Edge-IEConv  (Zhang et al., 2022)                  | w/o pretrain    | 0.8530       |
> > > | IEConv(Hermosilla et al., 2021)                           | w/o pretrain    | 0.8720       |
> > > | GearNet-Multiview-Contrast (Zhang et al., 2022)            | w/ pretrain     | 0.8750       |
> > > | New IEConv (Hermosilla and Ropinski, 2022)                        | w/ pretrain     | 0.8810       |
> > > | CDConv  (Fan et al., 2022)                        | w/o pretrain    | *0.8850*     |
> > > | **STELLA-ESM3-Llama-3.1-8B-Instruct**          | MMIT            |              |
> > > | - (single, two-stage, e3+e3)                   |                 | 0.8806       |
> > > | - (single, two-stage, e3+e6)                   |                 | **0.8885**   |

---

> > ### Author Response · Authors · 2024-11-27
> > **Response to Reviewer cosX (Follow-up concerns) -- #2 -- Regarding the model architecture and other aspects**
> >
> > **Response to follow-up concerns (regarding the model architecture and other aspects)**
> > - STELLA adopts a projection-based multimodal LLM architecture, inspired by widely recognized vision-language models like LLaVA[1]. Some recent works such as Prot2Text[2] and ProteinGPT[3] also unitize a projection layer (an MLP or a Linear layer) to build the multimodal framework. Similar efforts, such as ProtChatGPT[4] and 3D-MoLM[5], also draw from vision-language pretraining strategies such as the Q-former architecture in BLIP-2[6], to build the multimodal framework for biomolecular tasks. These models combine embeddings from a biomolecular modality (e.g., sequences or structures, or their fusion) with the text modality, aiming to leverage the capabilities of LLMs to handle biomolecular tasks.
> > - **The fundamental difference between STELLA and other previous models such as Prot2Text, ProteinGPT and ProtChatGPT is the fusion strategy for multiple modalities.** **These three previous models** typically integrate protein sequence and structure data using a **late fusion strategy**, where each modality is encoded separately before being fused. However, late fusion approaches have certain limitations, such as the potential loss of cross-modal relationships and increased complexity of encoder modules. In contrast, **STELLA adopts an early fusion strategy inherited from the ESM3 encoder**, where the sequence and structure modalities are jointly represented and fused into a unified representation at encoding stage. This early fusion strategy has the potential to both preserve the intrinsic relationships between modalities and improve computational efficiency.
> > > **[1]** Liu, H., Li, C., Wu, Q. and Lee, Y.J., 2024. Visual instruction tuning. Advances in neural information processing systems, 36.
> > > **[2]** Abdine, Hadi, et al. "Prot2text: Multimodal protein’s function generation with gnns and transformers." Proceedings of the AAAI Conference on Artificial Intelligence. Vol. 38. No. 10. 2024.
> > > **[3]** Xiao, Yijia, et al. "Proteingpt: Multimodal llm for protein property prediction and structure understanding." arXiv preprint arXiv:2408.11363 (2024).
> > > **[4]** Wang, C., Fan, H., Quan, R. and Yang, Y., 2024. Protchatgpt: Towards understanding proteins with large language models. arXiv preprint arXiv:2402.09649.
> > > **[5]** Li, S., Liu, Z., Luo, Y., Wang, X., He, X., Kawaguchi, K., Chua, T.S. and Tian, Q.,. Towards 3d molecule-text interpretation in language models. ICLR 2024.
> > > **[6]** Li, J., Li, D., Savarese, S. and Hoi, S., 2023, July. Blip-2: Bootstrapping language-image pre-training with frozen image encoders and large language models. In International conference on machine learning (pp. 19730-19742). PMLR.
> >
> > - **By inheriting the early fusion mechanism of ESM3, STELLA achieved SOTA performance in both functional description prediction and enzyme-catalyzed reaction prediction tasks under rigorous data splitting (sequence identity < 40% between the training and testing sets) and evaluation strategies with our curated OPI-Struc dataset**, demonstrating its successful application to the field of protein biology.

---

> > > ### Author Response · Authors · 2024-11-29
> > > **Additional Response to Reviewer cosX (Follow-up concerns)**
> > >
> > > Dear reviewer cosX,
> > >
> > > Thank you again for your thoughtful and constructive feedback on our work. We greatly appreciate your acknowledgment of certain aspects of our response to your initial concern. **Regarding the second concern you mentioned (i.e., the limited performance on major metrics, and the innovation of model architecture and other aspects), we have carefully addressed it, provided detailed explanations and revised the paper accordingly.** If our response has sufficiently addressed your concerns, we would be deeply grateful for any additional suggestions or feedback that could further enhance the quality of our paper. If there are still areas requiring further clarification, we would be more than happy to elaborate further.  **As the extended discussion phase is nearing its conclusion, we would greatly appreciate any feedback you might be able to share. Thank you for your time and expertise. Your insights are of great importance to the entire STELLA team, and we sincerely look forward to hearing your thoughts.**
> > >
> > > Best regards,
> > > STELLA Team
> > > ******
> > > For ease of reading, we summarize the main points of our response to your second concern as follows. More details can be found in the "Response to Reviewer cosX (Follow-up concerns) -- #1 and #2".
> > > - **STELLA achieved SOTA performance in both functional description prediction (FP) and enzyme-catalyzed reaction prediction (EP) task with ROUGE-L = 0.5257 and Accuracy = 0.8885**, surpassing previous respective benchmarks: Prot2Text_Large (ROUGE-L = 0.5140) for the FP task and CDConv (Accuracy = 0.8850) for the EP task, respectively.
> > > -  **As a novel multimodal LLM for protein biology, STELLA adopts an early fusion strategy inherited from the ESM3 encoder, different from previous multimodal LLMs, such as Prot2Text[1], ProteinGPT[2] and ProtChatGPT[3].** **These three models** typically integrate protein sequence and structure data using a **late fusion strategy**, where each modality is encoded separately before being fused. **In STELLA, the early fusion strategy make different modalities (i.e., sequence and structure) jointly represented and fused into a unified representation at encoding stage**, which has the potential to both preserve the intrinsic relationships between different modalities and improve computational efficiency.
> > > -  **STELLA adopts a stringent data splitting strategy to construct the OPI-Struc dataset, ensuring sequence similarity between training and test sets is below 40%.** This dataset split strategy better reflects real-world scenarios for annotating uncommon proteins. In doing so, **we present a very rigorous evaluation to assess STELLA's performance.**
> > >
> > > > [1] Abdine, Hadi, et al. "Prot2text: Multimodal protein’s function generation with gnns and transformers." Proceedings of the AAAI Conference on Artificial Intelligence. Vol. 38. No. 10. 2024.
> > > > [2] Xiao, Yijia, et al. "Proteingpt: Multimodal llm for protein property prediction and structure understanding." arXiv preprint arXiv:2408.11363 (2024).
> > > > [3] Wang, C., Fan, H., Quan, R. and Yang, Y., 2024. Protchatgpt: Towards understanding proteins with large language models. arXiv preprint arXiv:2402.09649.

---

### Official Review · Reviewer_XjeL · 2024-11-05

**Soundness:** 3
**Presentation:** 2
**Contribution:** 1
**Rating:** 3
**Confidence:** 5

**Summary:**

The paper introduces STELLA, a multimodal large language model (LLM) designed to enhance protein understanding by integrating structural representations with natural language processing. By enhancing LLM capabilities with protein structural information, STELLA aims to improve predictions of protein/enzyme functions. Comprehensive experiments were conducted to evaluate STELLA's performance on function prediction and enzyme name prediction tasks. The authors provide open access to the code and datasets for further research.

**Strengths:**

1. **Novelty**: STELLA presents a novel approach by combining protein structural data with LLMs.
2. **Open Access**: Open access to the code and datasets encourages collaboration and further innovation in the field.

**Weaknesses:**

1. **Usefulness**: Frankly, I don't find this work very useful practically. Usually, users who have the structure data of a protein already know its function. Even if not, they can use foldseek to find a list of structurally similar proteins, and infer its function from the annotations of these proteins (manually or using GPT-4o). The authors have not compared their method with this straightforward baseline.
2. **Technical contribution**: This work replaces the protein sequence encoder in existing work with a protein structure encoder. It reaffirms the superiority of ESM3 and Llama-3.1. Beyond that, I have not seen many significant technical contributions or insights that could inspire future work.

**Questions:**

1. Could you compare STELLA with the FoldSeek/Blastp + GPT-4o baseline, where the input of GPT-4o are the descriptions and e-values of the FoldSeek/Blastp-retrieved proteins?
2. How do you evaluate the accuracy on the EP task, considering enzymes can have alternative names? Here are a few examples: Lactase vs β-Galactosidase, Lipase vs Triacylglycerol Lipase, Catalase vs Hydrogen Peroxide Oxidoreductase, Alcohol Dehydrogenase vs ADH, Hexokinase vs ATP:D-hexose 6-phosphotransferase.
3. Could you reiterate the motivation of your work, i.e., what are the limitations of existing work and how STELLA contributed to the community?

---

> ### Author Response · Authors · 2024-11-24
> **Response to Reviewer XjeL-- #1**
>
> Thank you for raising 3 insightful questions for our work, which are addressed below.
>
>  **Response to Q1**
>
>  **(1) Methods**
> - Thank you so much for suggesting Foldseek as a baseline for comparison. Taking Foldseek as a straightforward baseline for comparison includes the following steps:
> > Step 1. **Structure retrieval**: Using Foldseek, we search the structure databases to identify structurally similar proteins with structural similarity to the target protein.
> > Step 2. **Function determination**: Functions of the retrieved proteins are annotated by mapping them to their corresponding entries in the Swiss-Prot database.
> - In our OPI-Struc dataset, the sequence similarity between the training and testing sets is below 40%, representing a challenging split that emulates real-world scenarios where models are required to generalize to predict functions for novel, unannotated proteins. In the first step, for the 4,203 protein structures in our test set, we used the Foldseek easy-search (https://github.com/steineggerlab/foldseek?tab=readme-ov-file#search) command with default parameters to search for similar protein structures within the training set for each test protein. For the e-value parameter, only matches with an e-value below 0.001 are considered and returned to the results. In the second step, the function prediction for proteins in the test set is assigned based on the functional annotation of the top 1 retrieved protein from the Swiss-Prot database. The median e-value of the top 1 retrieved proteins is 2.723e-20, indicating a high confidence in the retrieval results by Foldseek.
>
> **(2) Results**
> - Among the 4,203 protein structures in the test set, four structures did not have any similar matches in the training set identified by Foldseek. For the remaining 4,199 proteins, the ROUGE-L score achieved by Foldseek was 0.4586, which is significantly lower than the score of 0.5257 achieved by our model, STELLA-ESM3-Llama-3.1-8B-Instruct. For the four unfound proteins (Swiss-Prot/AFDB ID: P40532,O31983,P23487,Q15847), we double-checked using the Foldseek web interface (https://search.foldseek.com/search) and searched against the AFDB-SWISSPROT database, which contains detailed functional annotations. However, no similar structures were found for these proteins beyond the query structures themselves.
> - Here, we provide a specific example. A protein from the test set (Swiss-Prot ID: P14721) and a protein from the training set (Swiss-Prot ID: P51109) share identical functional descriptions but exhibit globally dissimilar structures (TM-score: 0.48). Our model produced a completely correct prediction with a ROUGE-L score of 1, whereas Foldseek delivered an entirely incorrect result with a ROUGE-L score of 0.
> - This highlights that our multimodal learning approach effectively captures more fine-grained relationships between protein structure and function, a capability that retrieval-based models inherently lack. The relationship between protein structure and function is highly complex, as demonstrated by a more detailed analysis of our dataset. Using Foldseek, we clustered the protein structures in the training set based on 90% structural similarity. Notably, only about 25% of proteins within the same cluster share identical functional annotations.
>
>  **Response to Q2**
> - Thank you for your question. To ensure a fair and rigorous comparison with other enzyme classification models, we converted the EC numbers to their unique official names according to the BRENDA Enzyme Database (https://www.brenda-enzymes.org/). In other words, each label corresponds to a single unique name. A model output is considered correct only if it fully matches the reference official enzyme names.

---

> ### Author Response · Authors · 2024-11-24
> **Response to Reviewer XjeL -- #2**
>
> **Response to Q3**
> - Protein biology revolves around the interplay of three data modalities: sequence, structure, and function (text).  Structural data  plays a vital role in uncovering protein biological functions. Although extensive structural data  including PDB and AFDB have been accumulated and publicly available, further efforts are needed to leverage these valuable resources for deeper understanding of protein biological functions. Fully realizing the potential of such plentiful structural data resource in protein understanding requires bridging the gap between structural data and the functional and biochemical insights essential for practical applications in both research and industry. To address this challenge, there is an urgent need to develop innovative approaches that integrate structural data with cutting-edge computational tools.
> - By establishing structural and functional correlations by utilizing condensed knowledge in LLMs, STELLA fulfils accurate, versatile and interactive predictions across a range of protein-related tasks (see the examples 1 and 2 below, which have also been put in Appendix A.1 in the revised manuscript). STELLA achieves state-of-the-art performance across multiple tasks, signalling the transformative potential of multimodal LLMs as an innovative paradigm in advancing protein biology research besides the development of protein language models (pLMs). The effective utilization of LLM capabilities has been recognized by Reviewer tXk1, cosX and L7ds as a strength of this work.
> - STELLA not only creates a new paradigm to understand proteins besides the development of pLMs, but also expands the boundaries of LLM capabilities in protein biology.  This study will help drive the advancement of protein science and computational biology through LLM-based approaches, establishing a new paradigm beyond the PLM-based paradigms.
>
> **Example 1**
> ```
> > User: What are the main functions of this protein?[AFDB_ID: AF-Q9W3K5-F1-model_v4.pdb]
> > STELLA: The main functions of this protein are to catalyze the ATP-dependent ligation of L-glutamate and L-cysteine, and to a lesser extent, of L-histidine and L-tryptophan.
> > User: Compare the function of glutamate-cysteine ligase with that of glutathione synthetase. What structural or mechanistic features distinguish these enzymes, and how do they together regulate glutathione levels?
> > STELLA: Glutamate-cysteine ligase and glutathione synthetase are both involved in the synthesis of glutathione, a critical antioxidant in cells. However, they have different functions and mechanisms of action. Glutamate-cysteine ligase catalyzes the ligation of L-glutamate and L-cysteine to form a peptide bond, while glutathione synthetase catalyzes the condensation of L-glutamate and L-cysteine to form glutathione. The two enzymes have different substrate specificities and reaction conditions, and they work together to regulate glutathione levels in cells.
> ```
> **Example 2**
> ```
> User: What are the main functions of this protein?
> [AFDB_ID: AF-Q5KYR2-F1-model_v4.pdb]
> > STELLA: The main functions of this protein are involved in the isomerization of 5-deoxy-glucuronate (5DG) to 5-dehydro-2-deoxy-D-gluconate (DKG or 2-deoxy-5-keto-D-gluconate).
> > User: Describe potential industrial or medical applications of the isomerization catalyzed by this protein, specifically converting 5-deoxy-glucuronate (5DG) to 5-dehydro-2-deoxy-D-gluconate (DKG). How might this reaction be utilized in biotechnology or pharmaceutical production?
> > STELLA: The isomerization reaction catalyzed by this protein has several potential applications in biotechnology and pharmaceutical production. For example, the conversion of 5DG to DKG could be used to produce novel compounds or to improve the efficiency of existing ones. Additionally, the reaction could be used to develop new biotechnological processes or to improve existing ones.
> > User: Any further examples?
> > STELLA: Yes, the isomerization reaction could also be used to produce novel antibiotics, anticancer agents, or other bioactive compounds. For example, the conversion of 5DG to DKG could be used to synthesize novel peptidoglycan structures, which are important for bacterial cell wall synthesis and could potentially be used to develop new antibiotics.
> ```

---

> > ### Comment · Reviewer_XjeL · 2024-11-26
> > **Response to the authors**
> >
> > Thank you for your efforts in revising the paper. Unfortunately, I believe the current version still lacks significant contribution and application potential required for acceptance.
> >
> > Btw, please note that the contribution of a paper should be concise and substantial. The things that this work has done and previous works had not should be clear.

---

> > > ### Author Response · Authors · 2024-11-30
> > > **Additional Response to Reviewer XjeL (Follow-up concerns)**
> > >
> > > Dear Reviewer XjeL,
> > >
> > > Thank you for your time and efforts in reviewing our work. We appreciate the opportunity to clarify the significant contribution of STELLA, which are as follows:
> > >
> > > - **We emphasize here the unique differences in model architecture and methodology between STELLA and previous works, including Prot2Text [1], ProteinGPT [2], and ProtChatGPT [3]**. While STELLA adopts a projection-based multimodal LLM architecture inspired by widely recognized vision-language models such as LLaVA [4], other models like Prot2Text and ProteinGPT also employ projection layers (e.g., MLP or Linear layers) to construct their multimodal frameworks, but with distinct protein encoders and LLM backbones. Similarly, ProtChatGPT and 3D-MoLM [5] draw from vision-language pretraining strategies, such as the Q-former architecture in BLIP-2 [6], to combine embeddings from biomolecular modalities (e.g., sequences or structures, or their fusion) with the text modality, leveraging LLMs for biomolecular tasks.
> > >
> > > - **The fundamental difference between STELLA and these prior models lies in their modality fusion strategy.** Previous models such as **Prot2Text, ProteinGPT, and ProtChatGPT typically adopt a late fusion strategy**, where protein sequence and structure data are encoded separately before being fused. However, late fusion approaches are often constrained by limitations such as the potential loss of cross-modal relationships and increased encoder complexity.
> > >
> > > - In contrast, **STELLA adopts an early fusion strategy, inherited from the ESM3 encoder**, wherein sequence and structure modalities are jointly represented and fused into a unified representation at the encoding stage. This early fusion approach not only preserves the intrinsic relationships between modalities but also enhances computational efficiency, offering a more streamlined and integrated solution for multimodal protein understanding.
> > > > [1] Abdine, Hadi, et al. "Prot2Text: Multimodal protein’s function generation with gnns and transformers." Proceedings of the AAAI Conference on Artificial Intelligence. Vol. 38. No. 10. 2024.
> > > > [2] Xiao, Yijia, et al. "ProteinGPT: Multimodal llm for protein property prediction and structure understanding." arXiv preprint arXiv:2408.11363 (2024).
> > > > [3] Wang, C., Fan, H., Quan, R. and Yang, Y., 2024. ProtchatGPT: Towards understanding proteins with large language models. arXiv preprint arXiv:2402.09649.
> > > > [4] Liu, H., Li, C., Wu, Q. and Lee, Y.J., 2024. Visual instruction tuning. Advances in neural information processing systems, 36.
> > > > [5] Li, S., Liu, Z., Luo, Y., Wang, X., He, X., Kawaguchi, K., Chua, T.S. and Tian, Q.,. Towards 3d molecule-text interpretation in language models. ICLR 2024.
> > > > [6] Li, J., Li, D., Savarese, S. and Hoi, S., 2023, July. BLIP-2: Bootstrapping language-image pre-training with frozen image encoders and large language models. In International conference on machine learning (pp. 19730-19742). PMLR.
> > >
> > > - **Besides, we want to highlight that STELLA achieves SOTA performance in two critical protein-related tasks**: functional description prediction (FP) and enzyme-catalyzed reaction prediction (EP). This was accomplished using a highly stringent data split strategy, ensuring sequence similarity between the training and test sets remains below 40%. While this strategy aligns with Prot2Text, it is significantly more rigorous than those used by ProteinGPT and ProtChatGPT. Such a stringent split better reflects real-world challenges in annotating uncommon or novel proteins, providing a robust and rigorous evaluation of STELLA’s performance.
> > >
> > > We sincerely hope to receive your additional feedback if our response has addressed your concerns. Thank you so much for your time and consideration.
> > >
> > > Best regards,
> > > STELLA Team

---

> > ### Author Response · Authors · 2024-11-26
> > **Response to Reviewer XjeL (Follow-up concerns) -- #1**
> >
> > - Thank you for raising this important concern. **We established a new paradigm based on multimodal LLMs, complementing traditional approaches based on protein language model**. Recently, leveraging LLMs to deal with biomolecular tasks has been attracting much research attention, such as ChatMol[1], 3D-MoLM[2], etc. STELLA opens a direction to apply multimodal LLMs to protein biology, aiming to providing tools for protein science research.
> > - On the one hand, STELLA **provides a new method to deal with downstream protein tasks with SOTA performance**. On the other hand, it **offers a new way to connect hidden knowledge of protein language and world knowledge of natural language**, which we think is of great significance in **discovering proteomics with the advancement of high-throughput sequencing in the future**. With accumulation of more and more proteins in life science, **there is a strong need to figure out their functions to further utilize these proteins in practical applications of life science**, such drug design and artificial enzyme design.
> > >[1] Zeng, Z., Yin, B., Wang, S., Liu, J., Yang, C., Yao, H., Sun, X., Sun, M., Xie, G. and Liu, Z., 2024. ChatMol: interactive molecular discovery with natural language. Bioinformatics, 40(9).
> > >[2] Li, S., Liu, Z., Luo, Y., Wang, X., He, X., Kawaguchi, K., Chua, T.S. and Tian, Q.,. Towards 3d molecule-text interpretation in language models. ICLR 2024.
> > - **As a new methodology, STELLA achieved SOTA performance in both the functional description prediction (FP) and enzyme-catalyzed reactions prediction (EP) tasks, with ROUGE-L = 0.5257 and accuracy = 0.8885**, respectively, surpassing previous respective benchmarks, including Prot2Text_Large (ROUGE-L = 0.5140) for the FP task and CDConv (Accuracy = 0.8850) for the EP task.
> >
> > > **Table: Evaluation results of the FP task, comparing with existing work.**  **Bold** and _italic_ indicate the best and the runner-up performance.
> >
> > >| Model/Method                            | BLEU-4 ↑ | BERT Score ↑ | ROUGE-1 ↑ | ROUGE-2 ↑ | ROUGE-L ↑ |
> > |-----------------------------------------|-----------|--------------|------------|------------|------------|
> > | Prot2Text$_{BASE}$ [abdine2023prot2text] | 0.3511    | 0.8430       | 0.5059     | 0.4271     | 0.4849     |
> > | Prot2Text$_{LARGE}$ [abdine2023prot2text] | 0.3629    | _0.8520_     | _0.5368_   | _0.4560_   | _0.5140_   |
> > | STELLA-ESM3-Llama-3.1-8B-Instruct (e3+e3) | _0.4024_  | 0.8496       | 0.5218     | 0.4487     | 0.5041     |
> > | STELLA-ESM3-Llama-3.1-8B-Instruct (e3+e6) | **0.4300**| **0.8564**   | **0.5423** | **0.4747** | **0.5257** |
> > | Foldseek + Manual retrieval            | 0.3627    | 0.8358       | 0.4799     | 0.4027     | 0.4586     |
> >
> > > **Table: Evaluation results of the EP task, comparing with existing work.**  **Bold** and *italic* indicate the best and runner-up performance.
> >
> > > | Model                                           | Training Manner | Acc@EP ↑     |
> > |------------------------------------------------|-----------------|--------------|
> > | DeepFRI (Gligorijevi´c et al., 2021)                              | w/ pretrain     | 0.6330       |
> > | UniRep (Alley et al., 2019)                              | w/o pretrain    | 0.7290       |
> > | 3DCNN (Derevyanko et al., 2018)                              | w/o pretrain    | 0.7880       |
> > | HH-suite3 (Steinegger et al., 2019)                           | w/o pretrain    | 0.8260       |
> > | ESM-1b (Rives et al., 2021)                            | w/ pretrain     | 0.8310       |
> > | GearNet-Edge-IEConv  (Zhang et al., 2022)                  | w/o pretrain    | 0.8530       |
> > | IEConv(Hermosilla et al., 2021)                           | w/o pretrain    | 0.8720       |
> > | GearNet-Multiview-Contrast (Zhang et al., 2022)            | w/ pretrain     | 0.8750       |
> > | New IEConv (Hermosilla and Ropinski, 2022)                        | w/ pretrain     | 0.8810       |
> > | CDConv  (Fan et al., 2022)                        | w/o pretrain    | *0.8850*     |
> > | **STELLA-ESM3-Llama-3.1-8B-Instruct**          | MMIT            |              |
> > | - (single, two-stage, e3+e3)                   |                 | 0.8806       |
> > | - (single, two-stage, e3+e6)                   |                 | **0.8885**   |

---

> > ### Author Response · Authors · 2024-11-26
> > **Response to Reviewer XjeL (Follow-up concerns) -- #2**
> >
> > - We acknowledge that the application potential of multimodal LLMs are a critical area of focus, not only in fields such as NLP and computer vision but also in the domain of protein biology. This aspect is one we deeply value and aim to contribute as much as possible.  We firmly believe that advancing the practical applications of such methodologies requires more long-term efforts. **STELLA not only achieved SOTA performance in two critical protein-related tasks but also push the frontier of LLM-based methodology in protein biology, which is a promising research direction.** The evolution of LLM-based methodologies for practical applications lies in progressively enhancing LLMs's capabilities, from knowledge retrieval to reasoning, and to task orchestration in an agent system. These advancements enable LLMs to tackle more complex, real-world application scenarios, such as the protein biology domain.

---

### Comment · Area_Chair_faFb · 2024-11-26
**responding to author rebuttals**

Dear reviewers, this paper has split reviews, and thus, we really need your help to engage. I would really appreciate it if you could share your responses to the author's rebuttal -- whether that solves your concern or not.

Thanks, AC vWrZ

---

### Meta-Review · Area_Chair_faFb · 2024-12-17

**Metareview:**

This paper proposes STELLA, a multimodal LLM-based approach that integrates protein structural data and LLM capabilities, achieving strong results on functional description and enzyme-catalyzed reaction tasks.

Initially, some reviewers were concerned about performance not surpassing SOTA methods (personally I do not feel beating sota is a critical criteria), but the authors provided an updated experiment (increasing stage-2 training epochs), leading to improved metrics surpassing established baselines. They also clarified architectural differences, emphasizing an early fusion strategy that differs from prior late-fusion approaches (e.g., Prot2Text), and more rigorously tested on a challenging data split with <40% sequence similarity.

The authors actively engaged with all reviewers (thank you!), addressing initial and follow-up concerns by adding experiments, examples, and clarifications on model training procedures. Not all reviewers rejoined the discussion after the last round of responses, but those who did acknowledge the improvements.

Some reviewers still questioned the novelty and practical utility, but the authors highlighted STELLA’s stronger performance after extended training and clarified their early fusion strategy. Another point raised by multiple reviewers was the comparison to simpler baselines like Foldseek + GPT; authors provided results showing STELLA outperformed such approaches. Also, queries about iterative feedback scenarios were given more examples, though the authors note that quantifying iterative improvements is still challenging.

I think this paper gets potential -- however, given it still requires so much improvement and change, I feel it is safer for the authors to fully refresh it for another submission.

**Additional Comments On Reviewer Discussion:**

see above

---

### Decision · Program_Chairs · 2025-01-22

Reject